# Gene flow mediates the role of sex chromosome meiotic drive during complex speciation

Colin D Meiklejohn[1]\*, Emily L Landeen[2†], Kathleen E Gordon[1‡], Thomas Rzatkiewicz[2], Sarah B Kingan[2§], Anthony J Geneva[2#], Jeffrey P Vedanayagam[2¶], Christina A Muirhead[2], Daniel Garrigan[2\*\*], David L Stern[3], Daven C Presgraves[2]\*

[1]School of Biological Sciences, University of Nebraska, Lincoln, United States; [2]Department of Biology, University of Rochester, New York, United States; [3]Janelia Research Campus, Howard Hughes Medical Institute, Virginia, United States

**\*For correspondence:**
cmeiklejohn2@unl.edu (CDM);
daven.presgraves@rochester.edu
(DCP)

**Present address:** [†]Department of Integrative Biology, University of California, Berkeley, Berkeley, United States; [‡]Department of Molecular Biology and Genetics, Field of Genetics, Genomics, and Development, Cornell University, New York, United States; [§]Pacific Biosciences of California, California, United States; [#]Department of Organismic and Evolutionary Biology, Harvard University, Cambridge, United States; [¶]Department of Developmental Biology, Sloan-Kettering Institute, New York, United States; [\*\*]AncestryDNA, San Francisco, United States

**Competing interests:** The authors declare that no competing interests exist.

**Abstract** During speciation, sex chromosomes often accumulate interspecific genetic incompatibilities faster than the rest of the genome. The drive theory posits that sex chromosomes are susceptible to recurrent bouts of meiotic drive and suppression, causing the evolutionary build-up of divergent cryptic sex-linked drive systems and, incidentally, genetic incompatibilities. To assess the role of drive during speciation, we combine high-resolution genetic mapping of X-linked hybrid male sterility with population genomics analyses of divergence and recent gene flow between the fruitfly species, *Drosophila mauritiana* and *D. simulans*. Our findings reveal a high density of genetic incompatibilities and a corresponding dearth of gene flow on the X chromosome. Surprisingly, we find that a known drive element recently migrated between species and, rather than contributing to interspecific divergence, caused a strong reduction in local sequence divergence, undermining the evolution of hybrid sterility. Gene flow can therefore mediate the effects of selfish genetic elements during speciation.
DOI: https://doi.org/10.7554/eLife.35468.001

## Introduction

Speciation involves the evolution of reproductive incompatibilities between diverging populations, including prezygotic incompatibilities that prevent the formation of hybrids and postzygotic incompatibilities that render hybrids sterile or inviable. Two patterns characterizing speciation implicate a special role for sex chromosomes in the evolution of postzygotic incompatibilities: Haldane's rule, the observation that hybrids of the heterogametic sex preferentially suffer sterility and inviability (*Haldane, 1922*; *Wu and Davis, 1993*; *Orr, 1997*; *Laurie, 1997*; *Price and Bouvier, 2002*; *Presgraves, 2002*; *Coyne and Orr, 2004*); and the large X-effect, the observation that the X chromosome has a disproportionately large effect on hybrid sterility (*Coyne and Orr, 1989*; *Coyne, 1992a*; *Presgraves, 2008*). These patterns hold across a wide range of taxa, including female heterogametic (*ZW*) birds and Lepidoptera and male heterogametic (*XY*) plants, *Drosophila*, and mammals (*Coyne and Orr, 1989*; *Coyne and Orr, 2004*). We now know that these 'two rules of speciation' (*Coyne and Orr, 1989*) are, in part, attributable to the rapid evolution of genetic factors that cause interspecific hybrid sterility on the X chromosome relative to the autosomes (*Tao and Hartl, 2003*; *Moehring et al., 2007*; *Masly and Presgraves, 2007*; *Presgraves, 2008*; *Good et al., 2008*). The relatively rapid accumulation of X-linked hybrid sterility factors is associated with reduced interspecific gene flow at X-linked *versus* autosomal loci (reviewed in *Presgraves, 2018*). Overall, these

patterns show that, for many taxa with heteromorphic sex chromosomes, the X chromosome plays a large and fundamental role in speciation.

Given the taxonomic breadth of Haldane's rule, the large X-effect, and reduced interspecific gene flow on the X, understanding *why* the X chromosome accumulates hybrid incompatibilities faster than the rest of the genome is imperative. At least five explanations have been proposed: faster X evolution (*Charlesworth et al., 1987*), gene traffic (*Moyle et al., 2010*), disrupted sex chromosome regulation in the germline (*Lifschytz and Lindsley, 1972*), the evolutionary origination of incompatibilities in parapatry (*Höllinger and Hermisson, 2017*), and meiotic drive (*Hurst and Pomiankowski, 1991*; *Frank, 1991*). Here, we focus on the potential role of meiotic drive. The drive theory posits that sex chromosomes are more susceptible than autosomes to invasion by selfish meiotic drive (*sensu lato*) elements (*Hurst and Pomiankowski, 1991*; *Frank, 1991*). Sex-linked drive compromises fertility and distorts sex ratios, which leads to evolutionary arms races between drivers, unlinked suppressors, and linked enhancers (*Lindholm et al., 2016*; *Presgraves, 2008*; *Meiklejohn and Tao, 2010*). These arms races can contribute to the evolution of hybrid male sterility, in at least two ways. Normally-suppressed drive elements might be aberrantly expressed in the naive genetic backgrounds of species hybrids, causing sterility rather than sex ratio distortion (*Hurst and Pomiankowski, 1991*; *Frank, 1991*). Alternatively, recurrent bouts of invasion, spread, and coevolution among drive, suppressor, and enhancer loci might cause interspecific divergence at these loci that incidentally cause hybrid sterility and map disproportionately to sex chromosomes (*Presgraves, 2008*; *Meiklejohn and Tao, 2010*).

Multiple lines of evidence support the plausibility of the drive theory. First, theoretical considerations and empirical evidence suggests that both active and suppressed sex chromosome meiotic drive systems are widespread in natural populations (*Jaenike, 2001*). Indeed, in one species, *Drosophila simulans*, three cryptic (normally suppressed) *sex-ratio* drive systems—Winters, Durham, and Paris—have been identified, involving distinct sets of X-linked drive loci and autosomal and/or Y-linked suppressors (*Tao et al., 2001*; *Tao et al., 2007a*; *Tao et al., 2007b*; *Helleu et al., 2016*). Second, loci involved in cryptic *sex-ratio* systems co-localize with hybrid male sterility loci in genetic mapping experiments (*Tao et al., 2001*; *Zhang et al., 2015*; *Orr and Irving, 2005*). Third, at least one of the two X-linked hybrid sterility genes identified to date also causes meiotic drive (*Phadnis and Orr, 2009*). These discoveries confirm that recurrent bouts of drive and suppression have occurred and that cryptic drive genes can cause hybrid sterility. While these findings put the plausibility of the drive hypothesis beyond doubt, the question of its generality remains: what fraction of X-linked hybrid sterility factors evolved as a consequence of drive? We can furthermore ask whether, and how often, drive can *impede* the evolution of hybrid incompatibilities. The drive hypothesis assumes, for instance, that populations evolve in strict allopatry (simple speciation) and/or that drive elements require particular population-specific genetic backgrounds for their activity. But for populations that diverge with some level of gene flow (complex speciation), drive elements can in principle migrate between species, thereby reducing divergence and potentially undermining the evolution of hybrid sterility (*Macaya-Sanz et al., 2011*; *Crespi and Nosil, 2013*; *Seehausen et al., 2014*).

Here, we investigate the special role of sex chromosomes in speciation with genetic mapping and population genomic analyses between *Drosophila mauritiana* and *D. simulans*. The human commensal species, *D. simulans*, originated on Madagascar, diverging from the sub-Saharan African species, *D. melanogaster*, ~3 Mya (*Lachaise et al., 1988*; *Dean and Ballard, 2004*; *Baudry et al., 2006*; *Kopp, 2006*; *Ballard, 2004*). The island-endemic species, *D. mauritiana*, originated on the Indian Ocean island of Mauritius, diverging from *D. simulans* ~240 kya (*Kliman et al., 2000*; *McDermott and Kliman, 2008*; *Garrigan et al., 2012*). The two species are now isolated by geography—*D. simulans* has never been collected on Mauritius (*David et al., 1989*)—and by multiple incomplete reproductive incompatibilities, including asymmetric premating isolation (*Coyne, 1992b*), postmating-prezygotic isolation (*Price, 1997*), and intrinsic postzygotic isolation ($F_1$ hybrid males are sterile, $F_1$ hybrid females are fertile; *Lachaise et al., 1986*). Despite geographic and reproductive isolation, there is clear evidence for historical gene flow between the two species (*Solignac and Monnerot, 1986*; *Solignac et al., 1986*; *Garrigan et al., 2012*; *Ballard, 2000a*; *Ballard, 2000b*; *Satta et al., 1988*; *Satta and Takahata, 1990*). The X chromosome shows both an excess of factors causing hybrid male sterility (*True et al., 1996b*; *Tao et al., 2003*) and, correspondingly, a dearth of historical interspecific introgression (*Garrigan et al., 2012*). The rapid

accumulation of X-linked hybrid male sterility factors may have contributed to reduced X-linked gene flow, limiting exchangeability at sterility factors and genetically linked loci (*Muirhead and Presgraves, 2016*).

To begin to assess the role of drive in the evolution of X-linked hybrid male sterility between these two species, we performed genetic mapping experiments using genotype-by-sequencing of advanced-generation recombinant X-linked introgressions from *D. mauritiana* in an otherwise pure *D. simulans* genetic background. In parallel, we performed population genomic analyses between *D. mauritiana* and *D. simulans* to study the chromosomal distributions of interspecific divergence and gene flow. These analyses lead to two discoveries regarding the role of meiotic drive in speciation. First, we find evidence for modest X-linked segregation distortion in hybrids, supporting the hypothesis that cryptic *sex-ratio* systems are common. Second, we show that a now-cryptic X-linked *sex-ratio* drive system recently introgressed between species and likely caused large selective sweeps in both species. As a result, this X-linked region shows greatly reduced interspecific sequence divergence and an associated lack of hybrid male sterility factors. Contra the drive hypothesis, in this instance, gene flow at a meiotic drive locus may have prevented or undermined the evolution of X-linked hybrid male sterility. These findings suggest that the effects of selfish genetic elements on interspecific divergence and the accumulation of incompatibilities depend on their opportunity to migrate between species during complex speciation.

## Results

### Mapping X-linked hybrid male sterility

Multiple intervals on the X chromosome cause male sterility when introduced from *D. mauritiana* into *D. simulans* (*True et al., 1996b*; *Maside et al., 1998*). The number and identities of the causal factors, how they disrupt spermatogenesis, and the evolutionary forces that drove their interspecific divergence are unknown. We therefore generated a high-resolution genetic map of X-linked hybrid male sterility between the two species, with the ultimate aim of identifying a panel of sterility factors. We first introgressed eight X-linked *D. mauritiana* segments that together tile across ~85% of the euchromatic length of the X chromosome into a *D. simulans* genetic background (*Figure 1A,B*; *Table 1*). Each introgressed segment was marked by two co-dominant P element insertions bearing mini-white transgenes (*P[w+]*; *True et al., 1996a*) that serve as visible genetic markers. We introgressed these '2P' segments into the *D. simulans* $w^{XD1}$ genetic background through >40 generations of repeated backcrossing (*Figure 1A*). Our ability to generate these introgression genotypes confirms that the distal 85% of the *D. mauritiana* X euchromatin carries no dominant factors that cause female sterility or lethality in a *D. simulans* genetic background (*True et al., 1996b*; *Tao et al., 2003*). All eight 2P introgression genotypes are, however, completely male-sterile, indicating that each of the introgressed regions contains one or more hybrid male sterility factors. Two pairs of introgression genotypes carry largely overlapping introgressed *D. mauritiana* segments and were combined for further analyses (2P-5a/b and 2P-6a/b, respectively; *Figure 1B*, *Table 1*).

To determine the genetic basis of male sterility within each 2P interval, we generated recombinant introgressions using *D. simulans* strains carrying *pBac[eYFP]* visible markers (*Stern et al., 2017*) (*Figure 1C*). These crosses capture unique recombination events between *P[w+]* and *pBac[eYFP]* markers, allowing recombinant *D. mauritiana* introgressions (hereafter called 1P-YFP) to be propagated indefinitely through females without recombination via selection for the 1P-YFP genotype. From these 1P-YFP females, an unlimited number of replicate males carrying identical 1P-YFP recombinant introgressions can be generated, assayed for male fertility, and archived for genotyping (*Figure 1C*; see below). We assayed male fertility in at least 10 individual males from each of 617 recombinant 1P-YFP genotypes (*Table 2*; see Materials and methods), and used the mean number of offspring across replicate males as the measure of fertility for each 1P-YFP genotype. Across 1P-YFP genotypes, the mean number of offspring ranged from 0 to 215 progeny; 238 genotypes (38.6%) were completely male-sterile, producing no offspring, and an additional 62 (10%) produced fewer than five offspring per male (*Figure 1—figure supplement 1*). Of the remaining 1P-YFP genotypes, 231 (37.4%) had intermediate fertility, and 86 (13.9%) had fertility indistinguishable from pure *D. simulans* controls ($P_{t-test}$ >0.01).

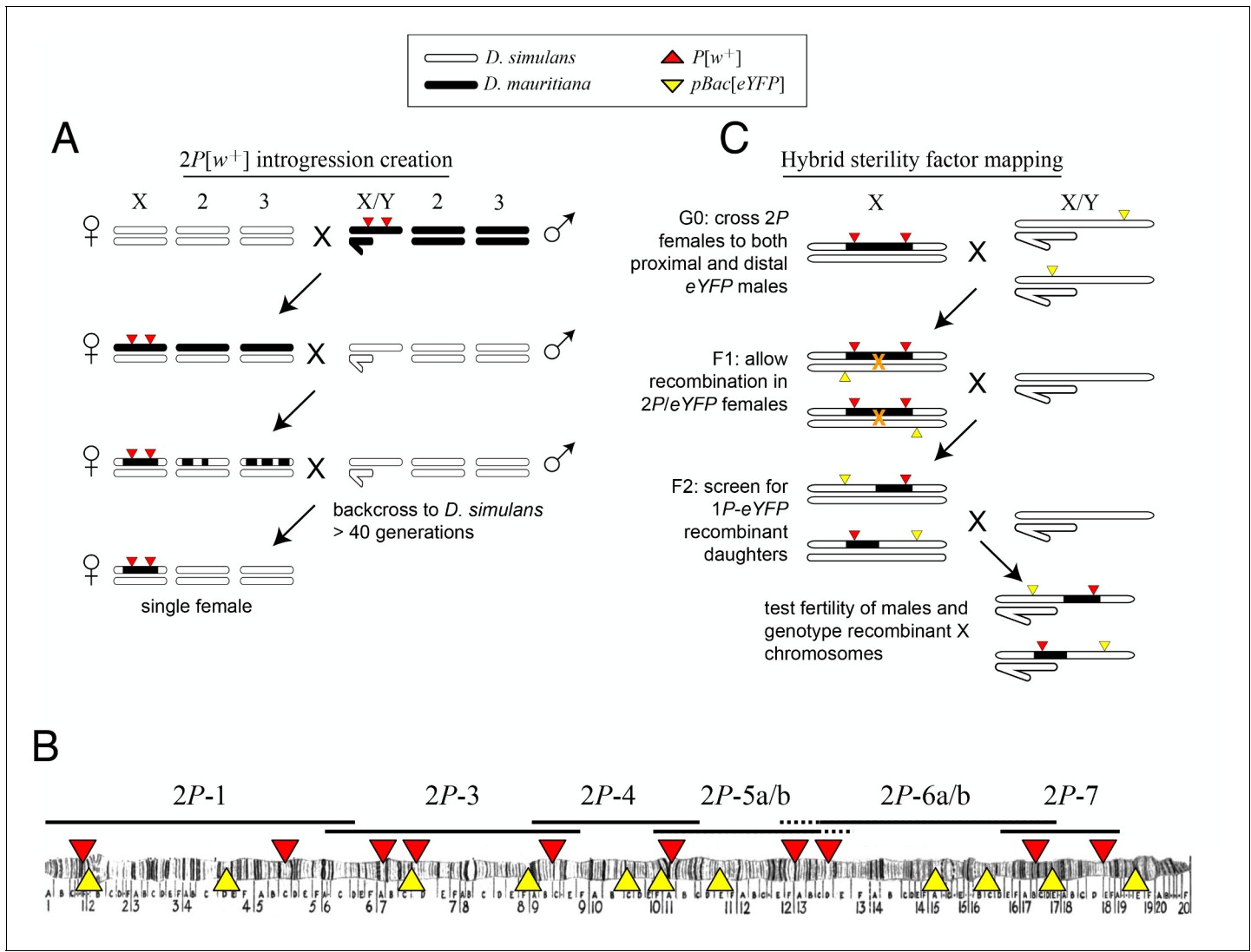

**Figure 1.** Crosses used to introgress eight regions of the *D. mauritiana* X chromosome into a *D. simulans* genome. (**A**) *D. mauritiana* '2P' lines were constructed by combining pairs of *P*-element insertions containing the miniwhite transgene (*P*[*w*⁺]; red triangles) distributed across the X chromosome. The *P*[*w*⁺] inserts are semi-dominant visible eye-color markers that permit discrimination of individuals carrying 0, 1 or 2*P*[*w*⁺]. X-linked segments from *D. mauritiana* were introgressed into a *D. simulans* genetic background by backcrossing 2*P*[*w*⁺] hybrid females to *D. simulans* *w*^XD1 males for over 40 generations. Each introgression line was then bottlenecked through a single female to eliminate segregating variation in the recombination breakpoints flanking the 2*P*[*w*⁺] interval. (**B**) Cytological map of the *D. melanogaster* X chromosome, indicating the locations of *P*[*w*⁺] and *pBac*[*eYFP*] transgene insertions. The extent of regions introgressed from *D. mauritiana* into *D. simulans* (e.g. 2*P*-1) are labeled above the map. Two pairs of introgression genotypes (2*P*-5a/b and 2*P*-6a/b) mostly overlap; the regions included in 2*P*-5b/2*P*-6b but not 2*P*-5a/2*P*-6a are indicated by dashed lines. (**C**) Meiotic mapping of sterility factors. 2*P*[*w*⁺] females were crossed to *D. simulans* strains carrying an X-linked *pBac*[*eYFP*] transgene (yellow triangles) that was used as an additional visible marker to score recombinant chromosomes. Recombinant X chromosomes with both *pBac*[*eYFP*] and a single *P*[*w*⁺] were chosen and assayed for male fertility. Recombinant chromosomes were generated using *pBac*[*eYFP*] markers both proximal and distal to each 2*P* introgression.

DOI: https://doi.org/10.7554/eLife.35468.002

The following source data and figure supplement are available for figure 1:

**Source data 1.** Source data for *Figure 1—figure supplement 1*, *Figure 4—figure supplement 1*.
DOI: https://doi.org/10.7554/eLife.35468.004
**Source data 2.** Source data for *Figure 1—figure supplement 1*, *Figure 4—figure supplement 1*.
DOI: https://doi.org/10.7554/eLife.35468.005
**Source data 3.** Source data for *Figure 1—figure supplement 1*.
DOI: https://doi.org/10.7554/eLife.35468.006

*Figure 1 continued on next page*

*Figure 1 continued*

**Figure supplement 1.** Distribution of fertility (number of progeny) among all males carrying recombinant 1P-YFP X chromosomes, and average number of progeny among all 1P-YFP genotypes.

DOI: https://doi.org/10.7554/eLife.35468.003

We determined high-resolution genotypes of 1P-YFP recombinant introgressions using multiplexed whole-genome sequencing (*Andolfatto et al., 2011*). After quality filtering, we obtained high-confidence genome-wide genotype information for 439 1P-YFP recombinant introgressions (*Figure 2*). No genotype showed evidence for any autosomal *D. mauritiana* alleles, confirming that the introgression scheme isolated X-linked *D. mauritiana* segments in a pure *D. simulans* autosomal genetic background (*Figure 2—figure supplement 1*). Recombinant 1P-YFP introgressions on the X chromosome ranged in size from 0.219 to 6.32 Mbp, with a mean length of 1.97 Mb (*Table 3*). *Figure 2* shows the distribution of *D. mauritiana* introgression segments and their corresponding sterility phenotypes. Three large regions on the *D. mauritiana* X chromosome can be introgressed into *D. simulans* without strong negative effects on male fertility, indicating an absence of major hybrid male sterility factors in these regions (*Figure 2*). Conversely, we delineated four small regions (<700 kb) that consistently and strongly reduced male fertility: 90% of replicate males with introgressions spanning these regions produce fewer than five offspring. Quantitative trait locus (QTL) analyses confirmed the existence of genetic variation among introgression genotypes that significantly affects male fertility (*Figure 3*, *Figure 3—figure supplement 1*). At least five QTL peaks are significant at $p < 0.01$ (permutation test). Most regions containing *D. mauritiana* alleles reduce the average number of progeny to <15. Two QTL peaks (2.5 cM, and 29.3 cM, *Figure 3*) appear to show higher fertility associated with the *D. mauritiana* allele than the *D. simulans* allele, but this is attributable to *D. mauritiana* sterility factors located at 12.6 cM and 17.5 cM and the negative linkage disequilibrium that is generated across a 2P interval by our meiotic mapping approach (*Figure 1C*).

## Sex ratio distortion revealed through experimental introgression

Among fertile 1P-YFP males, progeny sex ratios were skewed toward a slight excess of sons: the mean proportion of daughters was 0.45, and 86% of fertile 1P-YFP genotypes (260/303) produced fewer than 50% daughters (*Figure 4*). These skewed sex ratios are at least partially attributable to effects of the *sim w*[XD1] genetic background, as a similar male bias was observed among progeny of control *sim w*[XD1] males (mean proportion females = 0.46, $n = 35$ sires, $t$-test *vs.* null hypothesis of 0.5, p=0.005). We observe a significant positive correlation between fertility and progeny sex-ratio among both *sim w*[XD1] and introgression genotypes ($\rho = 0.44$, p=0.009; $\rho = 0.21$, p=0.0002, respectively); males that sire fewer progeny sire a lower proportion of daughters (*Figure 4—figure supplement 1*). However, there is some evidence that introgressed *D. mauritiana* alleles modify this modest male bias: across all fertile introgression genotypes, there is a significant negative correlation between the length of the introgressed *D. mauritiana* segment and the proportion of female progeny produced by that genotype ($\rho = -0.31$, p<0.0001, *Figure 4—figure supplement 2*). This

**Table 1.** Locations and lengths of 2P intervals.

| 2P interval | Left P[w+]* | Right P[w+]* | Length (Mbp) |
|---|---|---|---|
| 2P-1 | 993419 | 4498520 | 3.51 |
| 2P-3 | 6192555 | 9126133 | 2.93 |
| 2P-4 | 9126133 | 11189873 | 2.06 |
| 2P-5a | 11189873 | 13324017 | 2.13 |
| 2P-5b | 11189873 | 13903934 | 2.71 |
| 2P-6a | 13903934 | 17492084 | 3.59 |
| 2P-6b | 13324017 | 17492084 | 4.17 |
| 2P-7 | 17492084 | 18660037 | 1.17 |

*coordinate position in the assembled *D. simulans* $w^{501}$ genome

DOI: https://doi.org/10.7554/eLife.35468.007

**Table 2.** Fertility and sex ratio phenotypes for 1*P-YFP* recombinant genotypes.

| 2P interval | N tested | N sterile* | N sub-fertile | N fertile[†] | Mean fertility[†] | % fertile[†] | Mean SR[†] |
|---|---|---|---|---|---|---|---|
| 2P-1 | 171 | 48 | 20 | 103 | 72.2 | 0.60 | 0.43 |
| 2P-3 | 97 | 12 | 21 | 64 | 67.4 | 0.66 | 0.45 |
| 2P-4 | 77 | 17 | 9 | 51 | 71.9 | 0.66 | 0.45 |
| 2P-5a/b | 92 | 23 | 16 | 53 | 68.2 | 0.58 | 0.51 |
| 2P-6a/b | 97 | 69 | 10 | 18 | 73.8 | 0.19 | 0.44 |
| 2P-7 | 83 | 69 | 6 | 8 | 136.5 | 0.10 | 0.47 |
| all 1P-YFP genotypes | 617 | 238 | 82 | 297 | 81.7 | 0.48 | 0.45 |

*genotypes where no male produced any offspring

[†]genotypes where at least two males produced at least five offspring

DOI: https://doi.org/10.7554/eLife.35468.008

effect seems to be independent of the effects of introgressed alleles on fertility as the partial correlation between progeny sex-ratio and introgression length remains unchanged after taking into account the effect of fertility ($\rho = -0.31$, p<0.0001; *Figure 4—figure supplement 2*). One interpretation of these results is that the Y chromosome of *sim w*[XD1] causes weak segregation distortion, and the intensity of distortion is modified by X-linked alleles at multiple loci from *D. mauritiana*.

Although the majority of fertile 1*P-YFP* genotypes sired male-biased progeny, introgressions that included the distal end of the 2*P-5* region sired female-biased progeny (*Figure 4*). QTL analysis of progeny sex ratio confirms a significant peak in the distal portion of 2*P-5* (*Figure 4—figure supplement 3*). The estimated effect of this QTL on progeny sex ratios is 54.6% daughters for the *mauritiana* allele *versus* 42.5% daughters for the *simulans* allele. These results are consistent with the existence of a cryptic (normally-suppressed) X-linked drive allele in *D. mauritiana* that is released in a *D. simulans* genetic background, as the *D. mauritiana w*[12] strain used to generate the 2*P* introgressions produces slightly male-biased progeny sex-ratios using the same fertility assay (one male paired with three *D. simulans w*[XD1] females, n = 10 sires, mean sex-ratio = 0.47, *t*-test *vs. D. simulans w*[XD1]p=0.4). This region of the X chromosome does not contain any previously mapped meiotic drive loci in *D. simulans* (*Montchamp-Moreau et al., 2006*; *Tao et al., 2007a*; *Helleu et al., 2016*), suggesting that our experiments have uncovered a novel cryptic drive locus and provide the first evidence of cryptic X-chromosome drive in *D. mauritiana*.

## Population genomics of speciation history

The high density of hybrid male sterility factors and the presence of cryptic drive systems on the X chromosome is expected to influence patterns of gene flow between *D. mauritiana* and *D. simulans*. We therefore analyzed whole-genome variation within and between 10 *D. mauritiana* strains from Mauritius (*Garrigan et al., 2014*) and 20 *D. simulans* strains, including nine from Madagascar, ten from Kenya, and one from North America (*Rogers et al., 2014*; *Hu et al., 2013*). These data allow us to characterize differentiation and identify genomic regions with aberrant genealogical histories consistent with recent interspecific introgression. The analyses reported here complement earlier studies that characterized interspecific divergence (*Garrigan et al., 2012*), polymorphism within *D. mauritiana* (*Garrigan et al., 2014*; *Nolte et al., 2013*), and polymorphism within *D. simulans* (*Begun et al., 2007*; *Rogers et al., 2014*). Below we present genome-wide population genetic analyses using non-overlapping 10-kb windows (unless otherwise stated; see Materials and methods).

### Polymorphism

Our genome-wide analyses provide multiple indicators that the island-endemic *D. mauritiana* has a smaller effective population size than *D. simulans* (*Table 4*), consistent with previous multi-locus analyses (*Hey and Kliman, 1993*; *Kliman et al., 2000*). Compared to *D. simulans*, total polymorphism (*Nei and Li, 1979*) in *D. mauritiana* is 32% lower on the X chromosome and 19% lower on the autosomes (*Figure 5—figure supplement 1*). The X/autosome ratio of polymorphism is thus lower in *D. mauritiana* (0.656) than in *D. simulans* (0.778) and lower than the 3/4 expected for a random mating population with a 1:1 sex ratio (*Garrigan et al., 2014*). A substantial fraction of extant

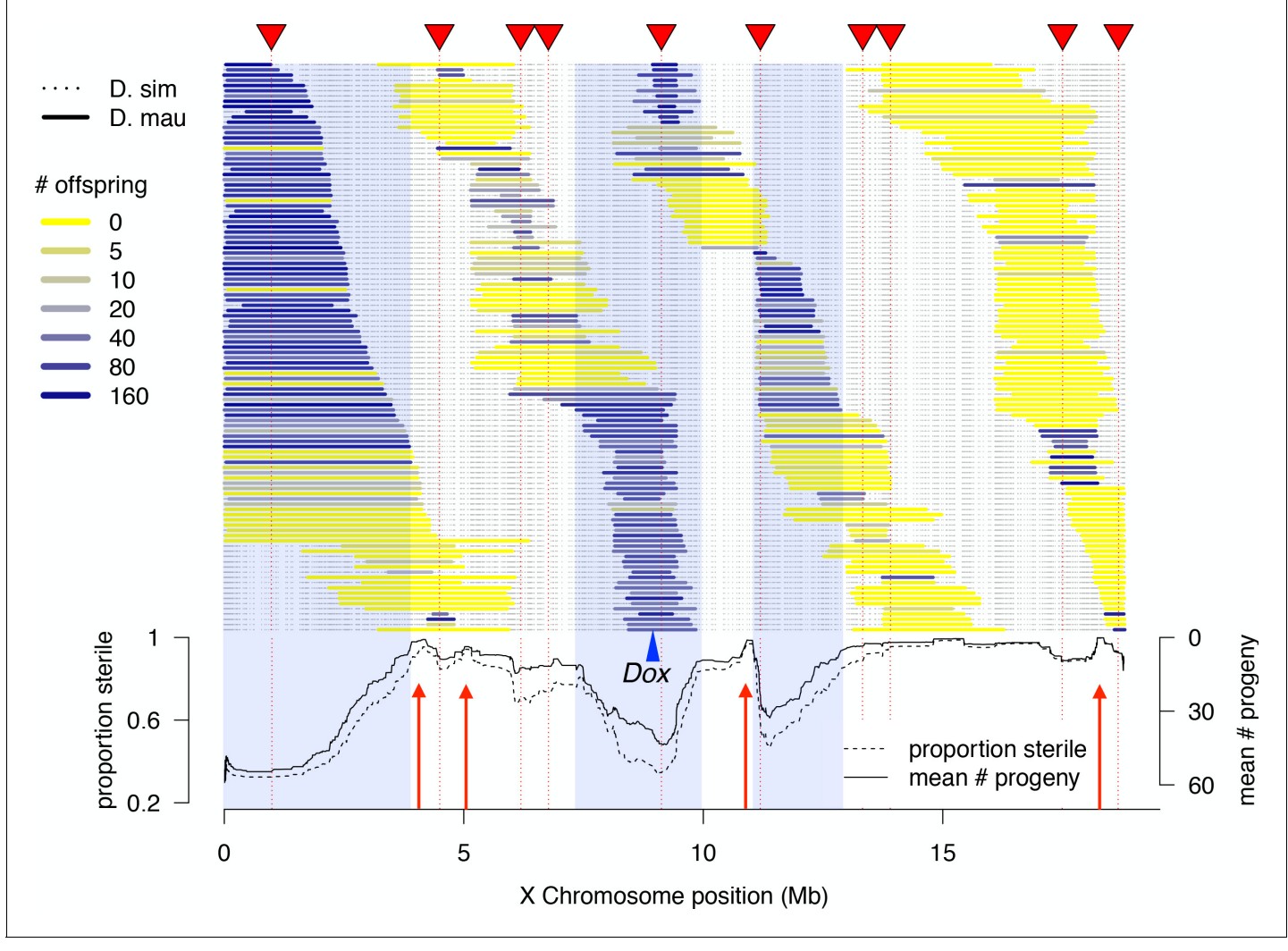

**Figure 2.** High-resolution genetic map of X-linked hybrid male sterility. Colored horizontal bars indicate the extent of introgressed *D. mauritiana* alleles for each recombinant 1*P-YFP* X chromosome. The color of each introgression indicates the mean fertility of 10 replicate males carrying that 1*P-YFP* X chromosome. The three shaded areas indicate fertile regions within which *D. mauritiana* introgressions do not cause sterility, whereas the four red arrows indicate small candidate sterility regions. The blue arrowhead indicates the location of the *Dox/MDox* meiotic drive loci. Lines in the lower panel indicate the average number of offspring and average proportion of sterile males (defined as producing fewer than five offspring) for all 1*P-YFP* genotypes that carry *D. mauritiana* alleles at each genotyped SNP.

DOI: https://doi.org/10.7554/eLife.35468.009

The following source data and figure supplement are available for figure 2:

**Source data 1.** Source data for *Figure 2*, *Figure 2—figure supplement 1*, *Figure 4*.
DOI: https://doi.org/10.7554/eLife.35468.011
**Figure supplement 1.** SNP locations and inferred ancestry for five recombinant 1*P-YFP* genotypes.
DOI: https://doi.org/10.7554/eLife.35468.010

polymorphisms in both species arose in their common ancestor, reflecting the large effective population sizes of both species and relatively recent species split time (see Materials and methods). Compared to *D. simulans*, however, *D. mauritiana* has retained 74.4% as many ancestral polymorphisms and accumulated just 46.3% as many derived polymorphisms. The site frequency spectra (*Tajima, 1989*) in *D. mauritiana* are less skewed toward rare variants than in *D. simulans*, and average linkage disequilibrium (*Kelly, 1997*) is twofold higher. Overall, these findings show that, relative to *D. simulans*, *D. mauritiana* has lower nucleotide diversity; retained fewer ancestral SNPs; accumulated fewer derived SNPs; a less negatively skewed site frequency spectrum; and greater linkage

**Table 3.** Distribution of 1P-YFP recombinant introgression lengths.

| 2P interval | Sequenced | Min size | Mean size | Max size |
| --- | --- | --- | --- | --- |
| 2P-1 | 129 | 295,225 | 2,617,833 | 6,322,871 |
| 2P-3 | 73 | 306,052 | 1,636,944 | 3,818,569 |
| 2P-4 | 55 | 226,018 | 1,482,659 | 2,917,578 |
| 2P-5 | 61 | 365,004 | 1,627,632 | 3,276,930 |
| 2P-6 | 55 | 692,350 | 2,400,499 | 4,764,204 |
| 2P-7 | 66 | 218,722 | 1,412,108 | 2,502,552 |

DOI: https://doi.org/10.7554/eLife.35468.016

disequilibrium—all patterns consistent with a historically smaller effective population size in *D. mauritiana* than in *D. simulans*.

## Divergence and differentiation

Net divergence levels between species are comparable to diversity levels within species. The median number of pairwise differences per site ($D_{XY}$) between the two species, estimated in non-overlapping 10-kb windows, is 0.010 for the X chromosome and 0.013 for the autosomes. However, as the X chromosome has lower levels of polymorphism within species, the median net divergence ($D_A$) between species is 0.0007 for the X (mean $D_A = 0.0007$) and $-0.0005$ (mean $D_A = -0.0006$) for the autosomes (a negative value of $D_A$ on the autosomes occurs because, on average, levels of within-species polymorphism exceed levels of between-species divergence). $D_A$ is significantly greater on the X chromosome than the autosomes (p<0.0001 for both medians and means). Allele frequency differentiation is also higher for the X chromosome (median $F_{ST} = 0.378$) than the autosomes (median $F_{ST} = 0.279$, $P_{MWU}$ <0.0001). These $F_{st}$ estimates imply that, for X-linked and autosomal loci, the mean times to coalescence for two gene copies sampled from the different species are 2.2- and 1.8-fold deeper than the mean coalescence times for two gene copies within-species, respectively (*Slatkin, 1993*).

## Recent interspecific gene flow and introgression

Gene flow between *D. mauritiana* and *D. simulans* has been rare during their speciation history, with an apparent recent increase (*Garrigan et al., 2012*). To identify genomic regions that have introgressed between species in the recent past, we used the $G_{min}$ statistic— the ratio of the minimum pairwise sequence distance between species to the average pairwise distance between species ($min[D_{XY}]/\overline{D}_{XY}$; *Geneva et al., 2015*). As populations diverge without gene flow, all loci in the genome gradually approach reciprocal monophyly, leaving just one ancestral lineage from each population available for coalescence in the ancestral population. Consequently, the minimum distance (numerator) equals the mean pairwise distance (denominator), causing $G_{min}\rightarrow 1$ with zero variance. Conversely, $G_{min}$ is small when the minimum distance is small relative to the mean pairwise distance. $G_{min}$ is therefore sensitive to genealogical configurations resulting from recent gene flow, particularly when introgressed haplotypes segregate at low to intermediate population frequency in at least one of the populations (*Geneva et al., 2015*). Importantly, $G_{min}$ distinguishes genealogies produced by introgression from those produced by incomplete lineage sorting. Between *D. mauritiana* and *D. simulans*, we find that median $G_{min}$ (±median absolute deviation) estimated for 10-kb windows across the major chromosome arms ranges from 0.761 ± 0.0537 for *3L* to 0.785 ± 0.0531 for the *X* (*Figure 5*; Kruskal-Wallis test, p<0.0001). As 95% of $G_{min}$ values are <0.85, reciprocal monophyly for 10-kb windows is rare.

To identify 10-kb outlier windows that have genealogical histories inconsistent with strict allopatric divergence, we used a Monte Carlo simulation procedure that assumes a constant species divergence time across all 10-kb intervals, separately for the X and the autosomes (see Materials and methods). In total, 196 of the 10,443 10-kb windows (1.9%) have a more recent common ancestry between *D. mauritiana* and *D. simulans* than expected under a strict allopatric divergence model, as indicated by significantly low values of $G_{min}$ ($P \leq 0.001$, corresponding to a genome-wide false discovery rate of 5%). As $G_{min}$ is a ratio, significantly small $G_{min}$ values could result from unusually small

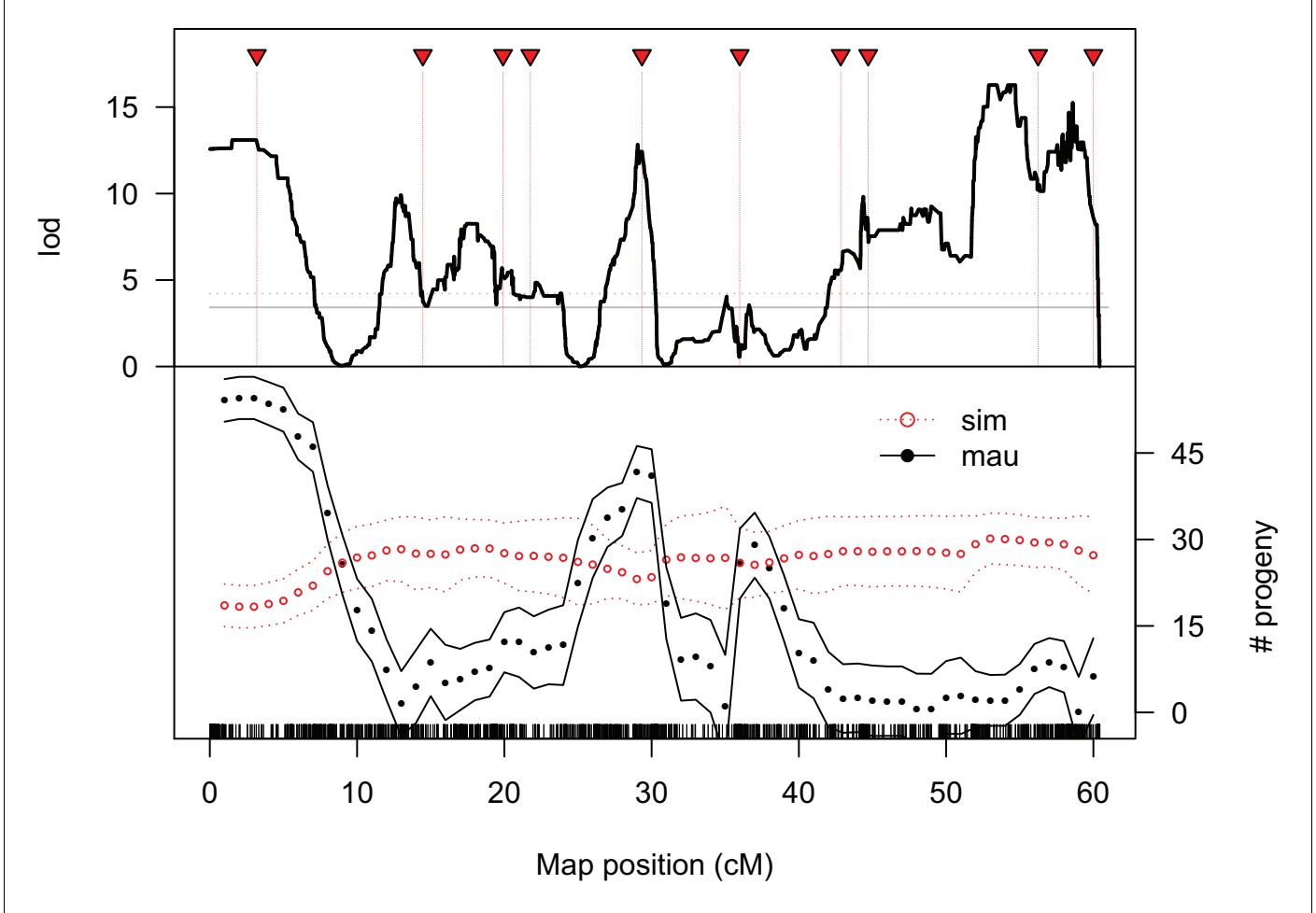

**Figure 3.** QTL analysis of male fertility. Mean offspring counts for each genotype were transformed as $\log_{10}(N + 1)$. The top plot shows lod scores for a two-part model that treats completely sterile genotypes as one class, and tests for quantitative effects on fertility among non-sterile genotypes. The solid and dotted gray lines indicate 5% and 1% significance thresholds, respectively, determined from 10,000 permutations. The bottom plot shows the estimated effects of *D. simulans* and *D. mauritiana* alleles at QTL placed every 1 cM (bounding lines indicate 95% confidence intervals).

DOI: https://doi.org/10.7554/eLife.35468.012

The following source data and figure supplements are available for figure 3:

**Source data 1.** Source data for *Figure 3*, *Figure 3—figure supplement 1*, *Figure 3—figure supplement 2*, *Figure 4—figure supplement 3*.
DOI: https://doi.org/10.7554/eLife.35468.015
**Figure supplement 1.** Alternate QTL models of male fertility.
DOI: https://doi.org/10.7554/eLife.35468.013
**Figure supplement 2.** QTL analysis of male fertility incorporating introgression length as a covariate.
DOI: https://doi.org/10.7554/eLife.35468.014

numerators (minimum $D_{XY}$) or unusually large denominators ($\overline{D}_{XY}$). We find that 10-kb windows with significant $G_{min}$ values have smaller median minimum $D_{XY}$ (0.0056 in introgression windows *versus* 0.0094 genome-wide, $P_{MWU} < 0.0001$) as well as *smaller* median $\overline{D}_{XY}$ (0.0110 in introgression windows *versus* 0.0124 genome-wide $P_{MWU} < 0.0001$), indicating that the significant $G_{min}$ values are due to unusually small minimum $D_{XY}$ values. The smaller $\overline{D}_{XY}$ of windows with significant $G_{min}$ reflects the contribution of the introgressed, low-distance haplotypes to the overall average pairwise distance between species.

Introgression windows are 4.4-fold underrepresented on the X chromosome: only nine of 1842 10-kb windows on the X chromosome (0.49%) have significant $G_{min}$ values *versus* 187 of 8601 10-kb windows on the autosomes (2.17%; Fisher's exact test p<0.0001). However, not all 10-kb

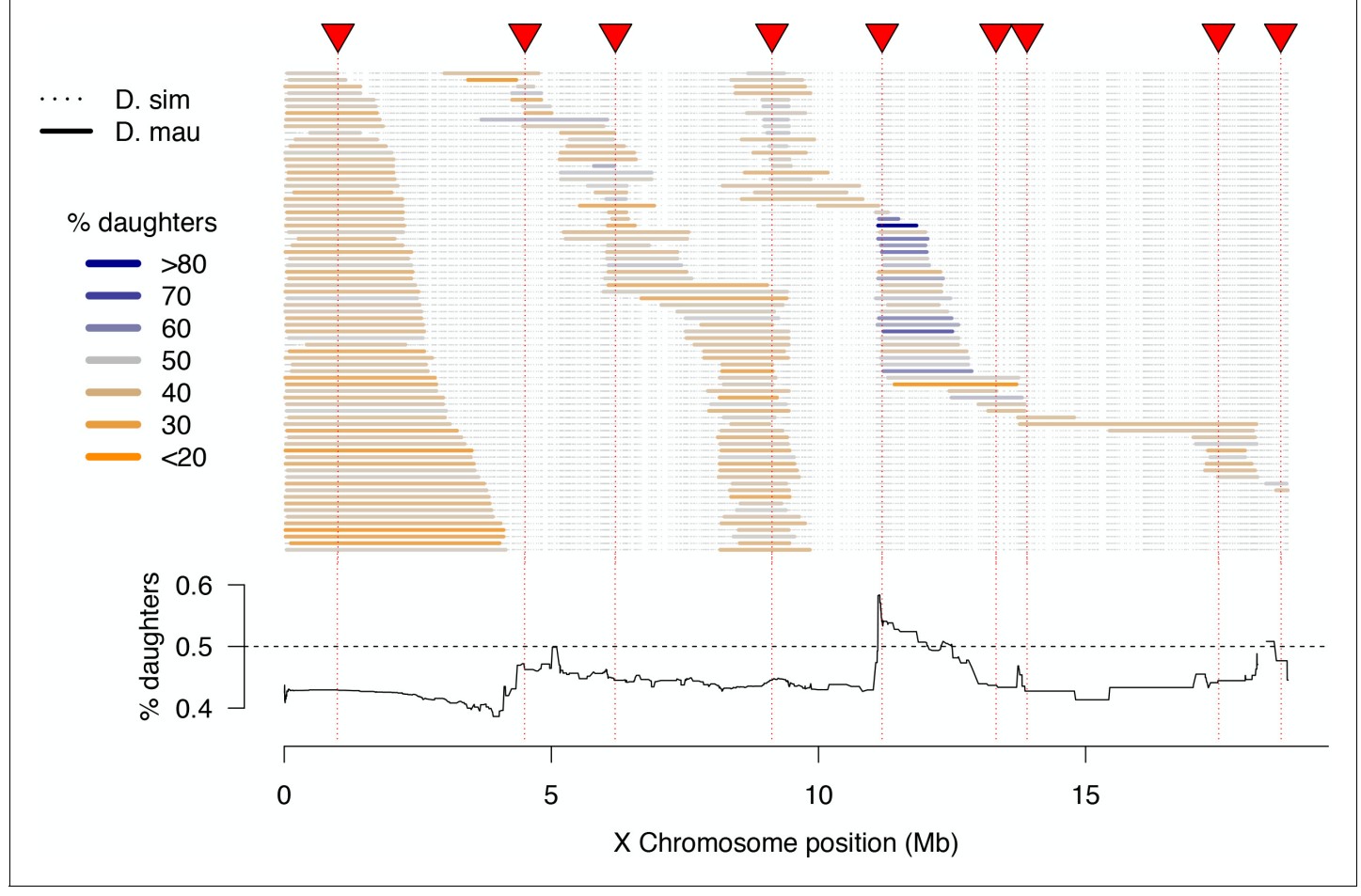

**Figure 4.** High-resolution map of progeny sex ratios among fertile 1*P-YFP* introgression male genotypes. Colored horizontal bars indicate the extent of introgressed *D. mauritiana* alleles for each fertile recombinant 1*P*-YFP X chromosome. The color of each introgression indicates the sex-ratio of progeny from replicate males carrying that 1*P*-YFP X chromosome. The line below indicates the average progeny sex-ratio for all 1*P*-YFP genotypes that carry *D. mauritiana* alleles at each genotyped SNP.

DOI: https://doi.org/10.7554/eLife.35468.017

The following figure supplements are available for figure 4:

**Figure supplement 1.** Relationship between progeny number and sex-ratio.

DOI: https://doi.org/10.7554/eLife.35468.018

**Figure supplement 2.** Relationship between introgression length, fertility, and sex-ratio.

DOI: https://doi.org/10.7554/eLife.35468.019

**Figure supplement 3.** QTL analysis of progeny sex ratio associated with introgression genotypes.

DOI: https://doi.org/10.7554/eLife.35468.020

introgression windows are independent: 169 of the 196 significant 10-kb windows (86.2%) can be arrayed into contiguous (or nearly contiguous) genomic regions (see Materials and methods). As a result, we infer 27 small (10-kb) introgressions and 21 larger introgressions ranging in size from 20 kb to 280 kb (*Supplementary file 1*). Of these 48 total introgressions, only one is on the X chromosome and 47 are on autosomes ($\chi^2$-test, p=0.0124). The lengths of these introgressed haplotypes depend on their time spent in the receiving population and on the local recombination rate. First, recombination has eroded introgression sizes over time, with longer, presumably younger, introgressions having smaller average $G_{min}$ values (Spearman $\rho = -0.6293$, p<0.0001) and smaller minimum $D_{xy}$ values ($\rho = -0.3677$, p=0.0101). Second, local recombination rate has been an important factor in determining introgression lengths, with relatively long introgressions tending to reside in chromosomal environments with low rates of crossing over ($\rho = -0.366$, p=0.0105).

**Table 4.** Population genomics summary statistics.

| Inference | Statistic* | D. simulans | D. mauritiana | P-value |
|---|---|---|---|---|
| Polymorphism | median $\pi_X$ | 0.0119 | 0.0076 | < 0.0001[‡] |
| | median $\pi_A$ | 0.0152 | 0.0116 | < 0.0001[‡] |
| | SNPs with inferred ancestry[†] | 4,324,740 | 2,181,959 | <0.0001[§] |
| | % ancestral SNPs | 14.6 | 21.6 | <0.0001[#] |
| | % derived SNPs | 85.3 | 78.3 | |
| Site frequency spectra | median Tajima's $D_X$ | −1.218 | −0.536 | < 0.0001[c] |
| | median Tajima's $D_A$ | −1.127 | −0.359 | < 0.0001[c] |
| Linkage disequilibrium | median $Z_{ns, X}$ | 0.056 | 0.122 | < 0.0001[c] |
| | median $Z_{ns, A}$ | 0.058 | 0.129 | < 0.0001[c] |

*Summary statistics estimated from 10-kb non-overlapping windows.

[†]SNP were inferred as ancestral or derived using parsimony, with *D. melanogaster* as an outgroup (see Materials and methods).

[‡]*P*-value for Mann-Whitney *U*-test.

[§]*P*-value for $\chi^2$-test.

[#]P-value from Fisher's exact test.

DOI: https://doi.org/10.7554/eLife.35468.021

To complement our distance-based $G_{min}$ analyses, we also used a genealogy-based four-population (ABBA-BABA) test, summarized by Patterson's *D*-statistic (*Green et al., 2010*; *Durand et al., 2011*), to evaluate the distribution of shared derived variants between *D. mauritiana* and *D. simulans*. Assuming a (((*D. sechellia*, *D. simulans*), *D. mauritiana*), *D. melanogaster*) tree topology, the null expectation is that a history involving zero gene flow should result in approximately equal numbers of ABBA and BABA nucleotide site configurations via lineage sorting, where A and B correspond to ancestral and derived states, respectively (*Green et al., 2010*; *Durand et al., 2011*). Instead, we find that *D* = 0.0812 (s.e. = 0.0033; block jackknife with 1 Mb blocks) across the genome, indicating a significant excess of shared derived sites between *D. simulans* and *D. mauritiana* compared to *D. sechellia* and *D. mauritiana*. These findings provide complementary support for a history of interspecific gene flow between *D. mauritiana* and *D. simulans*.

## Interspecific introgression of the cryptic Winters *sex-ratio* drive system

The single introgression detected on the X chromosome corresponds to a ~130-kb region that comprises eight protein-coding genes plus the Winters sex-ratio meiotic drive genes, *Distorter on the X* (*Dox*) and, its progenitor gene, *Mother of Dox* (*MDox*) (*Tao et al., 2007a*) (*Figure 6*). The median $G_{min}$ value across this 130-kb region is 0.333, a ~2.4-fold reduction relative to background $G_{min}$ on the X chromosome ($P_{MWU}$ <0.0001). The most extreme 10-kb window within the 130-kb region has a minimum $D_{XY}$ value (=0.00087) that is 92% smaller than the X chromosome-wide $\overline{D}_{XY}$, implying that introgression occurred in the recent past. The 130-kb region is also an outlier with respect to Patterson's *D* statistic: we observe 90.2 (72%) ABBA sites *versus* just 35.2 (28%) BABA sites in the region (*D* = 0.4382), whereas a significantly different configuration of ABBA and BABA sites occurs on the X chromosome outside the 130-kb region (9774.6 [55%] and 7911.1 [45%], respectively; *D* = 0.1054; $\chi^2$-test, p=0.00027). The elevated value of *D* within the 130-kb region indicates a significant excess of derived nucleotide variants shared between *D. simulans* and *D. mauritiana* compared to genomic background levels. Given the evidence from both distance- and genealogy-based analyses, we conclude that this 130-kb haplotype has a history of recent gene flow between species. In *D. simulans*, when unsuppressed, *MDox* and *Dox* cause biased transmission of the X chromosome during spermatogenesis, with male carriers siring more than 80% daughters (*Tao et al., 2007a*). These drivers are suppressed by an autosomal gene, *Not much yin* (*Nmy*), a retrotransposed copy of *Dox* that is a source of endogenous siRNAs that silence both *MDox* and *Dox* (*Tao et al., 2007b*). In non-African *D. simulans* populations, *Dox*, *MDox*, and *Nmy* are nearly fixed, although haplotypes lacking functional copies of the genes segregate at low frequencies (*Kingan et al., 2010*). All three loci have histories consistent with selective sweeps in multiple populations of *D. simulans* due to the presumed

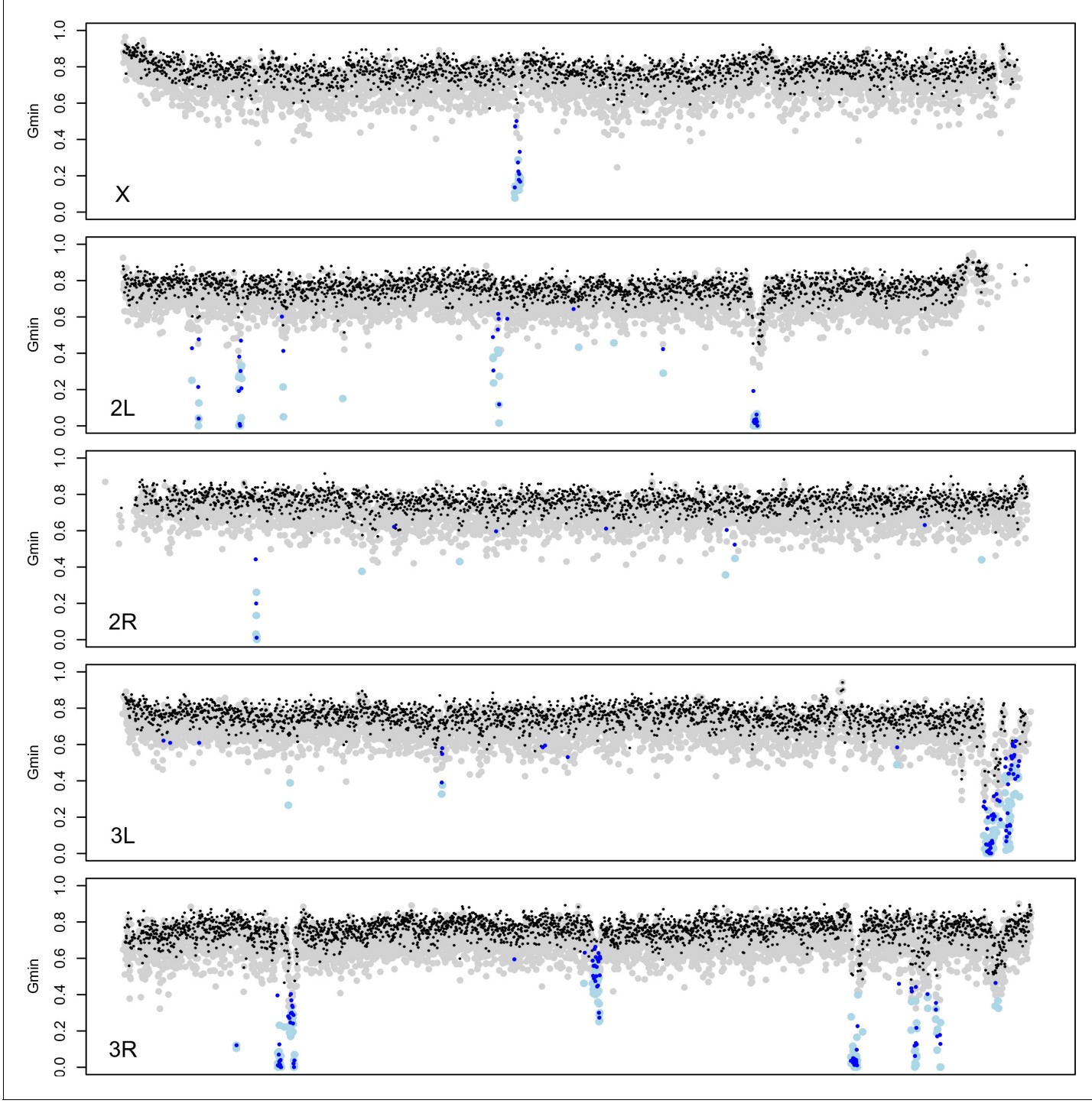

**Figure 5.** Identification of introgessed regions by $G_{min}$. Grey (black) dots indicate $G_{min}$ values calculated using 5-kb (10-kb) windows; light blue (dark blue) dots indicate 5-kb (10-kb) windows with significant $G_{min}$ values. As with 10-kb windows, 5-kb windows with significant $G_{min}$ values are 4-fold underrepresented on the X chromosome: 14 of 3603 5-kb windows on the X chromosome (0.39%) have significant $G_{min}$ values *versus* 266 of 17,065 5-kb windows on the autosomes (1.56%; Fisher's exact test p<0.0001).

DOI: https://doi.org/10.7554/eLife.35468.022

The following source data and figure supplements are available for figure 5:

**Source data 1.** Source data for *Figure 5—figure supplements 1* and *2*.
DOI: https://doi.org/10.7554/eLife.35468.025

**Source data 2.** Source data for *Figure 5*.

*Figure 5 continued on next page*

*Figure 5 continued*

DOI: https://doi.org/10.7554/eLife.35468.026

**Figure supplement 1.** Population genomic scans for polymorphism, divergence, and introgression in 10-kb windows.

DOI: https://doi.org/10.7554/eLife.35468.023

**Figure supplement 2.** Polymorphism and $G_{min}$.

DOI: https://doi.org/10.7554/eLife.35468.024

transmission advantage at *MDox* and *Dox* and the associated selective advantages of suppressing drive and restoring equal sex ratios at *Nmy* (*Kingan et al., 2010*). We estimated the probability that a random X-linked 130-kb introgression might include *Dox* and *MDox* by chance by permuting the location of a 130-kb segment on the X chromosome. Out of 100,000 such random permutations, 356 included *Dox* and *MDox* (p=0.004). We hypothesize that the signature of recent introgression at these sex-ratio distorters is not coincidental, but rather that introgression was mediated by their biased transmission through males.

Maximum-likelihood phylogenetic trees for the 130-kb *MDox-Dox* region show reduced diversity within *D. mauritiana* and reduced divergence between the two species (*Figure 6*). Among the 10 *D. mauritiana* sequences, nucleotide diversity is just 24% ($\pi$ = 0.0018) of background diversity levels on the X chromosome, corresponding to a massive selective sweep in the *D. mauritiana* genome ($P_{MWU}$ <0.0001; see also (*Nolte et al., 2013*; *Garrigan et al., 2014*)). The distribution of variability among haplotypes in the *D. simulans* samples is consistent with a parallel, albeit incomplete, selective sweep (*Figure 6*).

To determine if the *MDox* and/or *Dox* drive elements are associated with introgression between species and the selective sweeps within each species, we determined *MDox* and *Dox* presence/absence status for each line using diagnostic restriction digests (see Materials and methods). In contrast to previous work showing that *MDox* and *Dox* are nearly fixed among *D. simulans* samples collected outside of Africa (*Kingan et al., 2010*), we find that the drivers are at lower frequency among our 19 African samples (9 Madagascar, 10 Kenya): five have *MDox* (26%), five have *Dox* (26%), and only one has both genes (5%; NS33; *Supplementary file 2*). Despite these low frequencies, *MDox* and *Dox* are overrepresented among the haplotypes shared between species: 6 of the 7 shared haplotypes have *MDox* and/or *Dox* (Fisher's Exact $P_{FET}$ = 0.0018), and 2 of the 7 possess both drivers ($P_{FET}$ = 0.0158; $n$ = 19 African samples, plus the reference strain, *D. simulans* $w^{501}$, which has both). In *D. mauritiana*, all 10 lines have *MDox*, but only two have *Dox* (*Figure 6*; *Supplementary file 2*). RT-PCR shows that *MDox* is expressed in testes from both species (see Materials and methods), confirming its potential activity. These findings provide support for the hypothesis that segregation distortion mediated by *Dox* and (transcriptionally active) *MDox* genes was responsible for introgression and the parallel sweeps at this locus.

Notably, the large *MDox-Dox* introgression, and its associated sweep co-localize with one of the three regions of the X chromosome that, in our mapping experiments, fails to cause male sterility when introgressed from *D. mauritiana* into *D. simulans* (*Figure 2*). These observations suggest that a driving haplotype moved between species and swept to high frequency in *D. simulans* and fixation in *D. mauritiana*, thereby reducing local sequence divergence between species. This discovery has two implications. First, the *MDox-Dox* region is the only locus on the X chromosome to have recently escaped from its linked hybrid incompatibility factors and introgressed between species. Second, by sweeping to high frequency or fixation, the *MDox-Dox* drive element region reduced local divergence between species and, incidentally, undermined the accumulation of genetic incompatibilities that might cause hybrid male sterility.

## Discussion

Our combined genetic and population genomics analysis of hybrid male sterility and gene flow between *D. mauritiana* and *D. simulans* yields three findings. First, we confirm the rapid accumulation of X-linked hybrid male sterility between these species and map four major sterility factors to small (<700 kb) intervals (*Figure 2*). Second, we find that very recent natural introgression has occurred between these species, albeit almost exclusively on the autosomes, consistent with a large X-effect on gene flow (*Supplementary file 1*). Third, we discover new roles for meiotic drive during

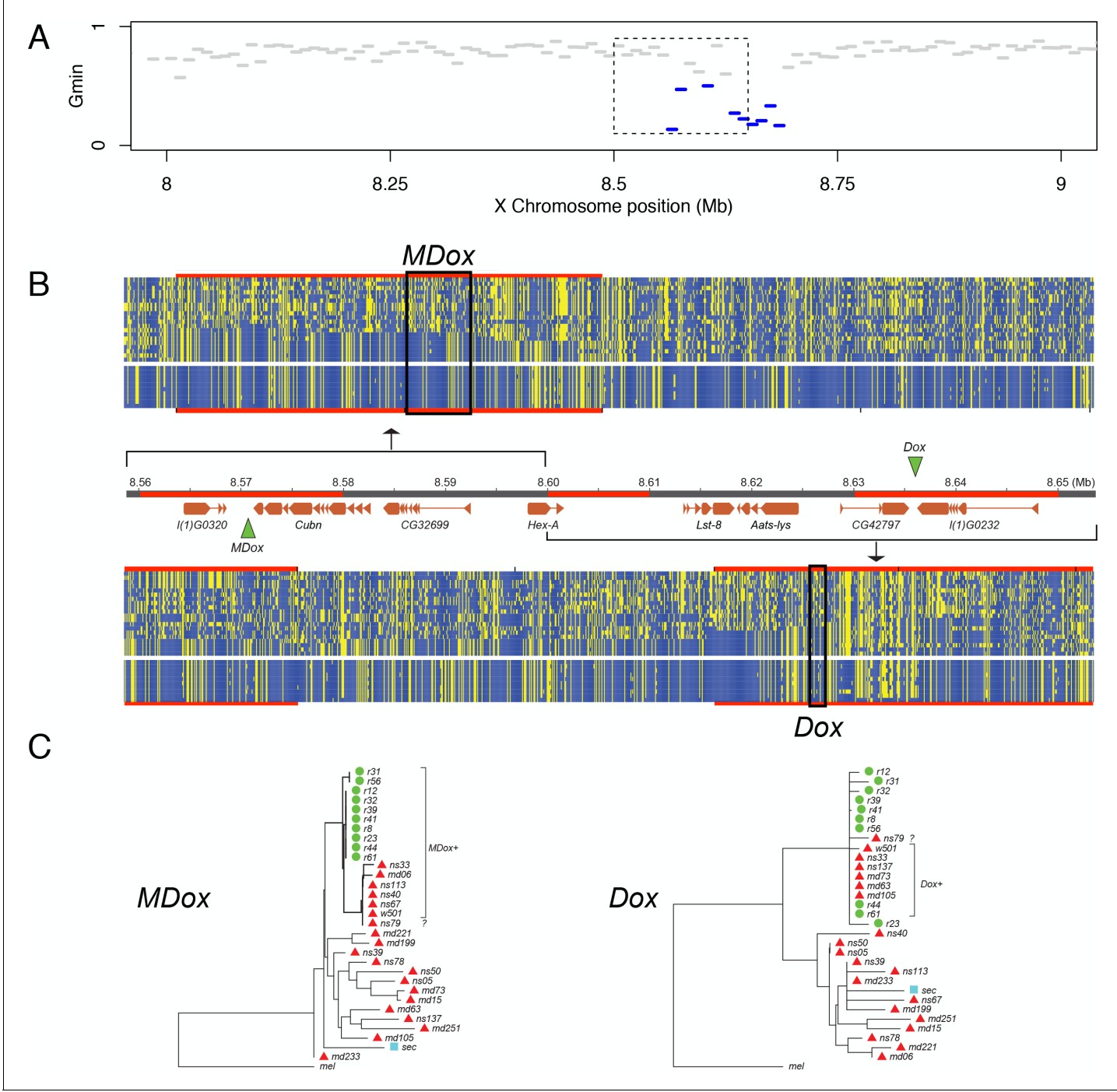

**Figure 6.** Natural introgression of the *MDox-Dox* region of the X chromosome. (**A**) $G_{min}$ values for 10-kb windows in the region containing *MDox* and *Dox*. Blue lines indicate windows with significantly low $G_{min}$ values. Inset box indicates the 90-kb region shown in panel B. (**B**) DNA polymorphism tables: the top table corresponds to the *MDox* region, and the bottom corresponds to the *Dox* region. Within the tables, yellow squares denote the derived nucleotide state, and blue squares indicate the ancestral state. The top 20 rows of each table correspond to the *D. simulans* samples, and the bottom 10 rows correspond to the *D. mauritiana* samples. The genome map between the polymorphism tables shows gene models for the region (orange boxes) and the locations of the *MDox* and *Dox* genes (green triangles). Regions highlighted in red are 10-kb windows with significantly low $G_{min}$ values. (**C**) Maximum likelihood phylogenetic trees for the *MDox* and *Dox* regions. Green circles and red triangles denote *D. mauritiana* and *D. simulans* samples, respectively.

DOI: https://doi.org/10.7554/eLife.35468.027

The following source data is available for figure 6:

*Figure 6 continued on next page*

*Figure 6 continued*

**Source data 1.** Source data for *Figure 6*.
DOI: https://doi.org/10.7554/eLife.35468.028

the history of speciation between these species. Some drive seems to be associated with functional divergence between species: one region of the *D. mauritiana* X chromosome appears to cause segregation distortion in a *D. simulans* genetic background. In contrast, the well-characterized X-linked Winters *sex ratio* distorters, *MDox* and *Dox*, have clearly migrated between species, reducing local interspecific divergence. Together, these findings, respectively, suggest that genetic conflict may both promote as well as undermine the special role of sex chromosomes in speciation.

## Genetic basis of X-linked hybrid male sterility

Our genetic analyses were initiated by introgression of six different regions of the *D. mauritiana* X chromosome into a pure *D. simulans* genetic background. All six regions cause complete hybrid male sterility and therefore carry at least one, or a combination of, *D. mauritiana* allele(s) that disrupt spermatogenesis due to incompatibilities with X-linked, Y-linked, or autosomal *D. simulans* alleles. Only three large (>2 Mb) regions of the *D. mauritiana* X are readily exchangeable between species, permitting male fertility in a *D. simulans* genome. Thus, after only ~250,000 years, sufficient X-linked hybrid male sterility has accumulated to render most of the *D. mauritiana* X chromosome male-sterile on a *D. simulans* genetic background (*True et al., 1996b*). Most of the *D. mauritiana* X chromosome is male-sterile in a *D. sechellia* genome as well (*Masly and Presgraves, 2007*). The combination of such extensive reproductive isolation with such modest genetic divergence makes this species group an ideal system to study the genetic basis of speciation.

We were able to define four small regions (<700 kb), each sufficient to cause complete male sterility (*Figure 2*), suggesting that these may contain single, strong sterility factors. We also find a large region spanning most of 2*P*-6 from which we were unable to recover fertile 1*P-YFP* recombinants. We infer that 2*P*-6 contains a minimum of two strong sterility regions, one tightly linked to each of the flanking *P*-elements (*Figure 3*). While our 2*P* mapping scheme is designed to facilitate the identification of male sterility factors, the 2*P*-6 interval highlights one of its limitations: in regions like 2*P*-6, for which strong sterility factors are very close to both flanking *P*-elements, we cannot determine how many additional sterility factors might localize to the middle of the interval. The present experiments therefore provide only a minimum estimate of the total number of hybrid male sterility factors on the X chromosome. We tentatively conclude that, within the fraction of the *D. mauritiana* X chromosome investigated, there are at least six genetically separable regions, each individually sufficient to cause virtually complete male sterility. It is worth noting that these experimental approaches detect relatively large-effect sterility factors under a single set of laboratory conditions. There are likely many hybrid male sterility factors of smaller effect, generally neglected in the lab but easily detected by selection in natural populations and thus able to affect the probability of migration at linked loci.

## Genomic signatures of complex speciation with gene flow

The two species studied here are allopatric: *D. simulans* has never been reported on Mauritius, and *D. mauritiana* has never been found anywhere other than Mauritius (*David et al., 1989*; *Legrand et al., 2011*). *D. mauritiana* appears to have originated from a *D. simulans*-like ancestor, probably from Madagascar, that migrated and established a population on Mauritius (*Hey and Kliman, 1993*; *Kliman et al., 2000*). Our characterization of genome-wide variation within and between *D. mauritiana* and *D. simulans* confirms a coalescent history that reaches considerably deeper into the past than the inferred species split time of ~250,000 years (*Hey and Kliman, 1993*; *Kliman et al., 2000*). Nested within this largely shared coalescent history, many functional differences have evolved between the two species, including extreme ones that mediate large-effect hybrid incompatibilities. The signatures of gene flow found in the genomes of these species imply recurrent bouts of migration and interbreeding. To introgress between species, immigrating foreign haplotypes must escape their locally disfavored chromosomal backgrounds by recombination before being eliminated by selection against linked incompatibilities and locally maladaptive alleles

(*Petry, 1983*; *Bengtsson, 1985*; *Barton and Bengtsson, 1986*). Conditional on escape, the lengths of foreign haplotypes will be subject to gradual erosion by recombination with the resident genetic background.

Here, and in previous work (*Garrigan et al., 2012*), we detect evidence consistent with weak migration: 2–5% of the genome shows evidence of introgression between *D. simulans* and *D. mauritiana* during their recent history. Our population genomic analysis identified 48 segregating foreign haplotypes. We find evidence that the genomic locations and lengths of introgressed foreign haplotypes have been shaped by selection and by recombination in the receiving population. First, selection has likely affected the genomic distribution of foreign haplotypes: only one of the 48 introgressions occurs on the X chromosome. The opportunity for foreign haplotypes on the X chromosome to escape linked incompatibilities via recombination is more constrained than on the autosomes, as the X has a higher density of incompatible alleles, and hemizygous selection eliminates foreign X-linked haplotypes more quickly (*Muirhead and Presgraves, 2016*). Second, we find that the lengths of introgressed haplotypes depend on local recombination rates: introgressions tend to be longer in chromosomal regions with relatively lower recombination rates. Third, after escaping locally deleterious chromosomal backgrounds, recombination eroded the lengths of foreign haplotypes over time: recently introgressed, and hence less diverged, haplotypes tend to be longer. It is worth noting here that the 10-kb windows used for our $G_{min}$ scan for foreign haplotypes almost certainly fails to identify very small and/or old introgressions. However, similar results are obtained from $G_{min}$ scans using 10-kb and 5 kb windows (*Figure 5*).

## Meiotic drive and complex speciation

The original drive theory posits that hybrid incompatibilities accumulate as incidental by-products of recurrent bouts of meiotic drive and suppression (*Hurst and Pomiankowski, 1991*; *Frank, 1991*). Our mapping experiments provide no direct evidence in support of this theory in *D. mauritiana* and *D. simulans*, as no hybrid male sterility loci co-localized with sex-ratio loci. Direct genetic evidence that sex-ratio distortion is responsible for the evolution of hybrid male sterility is however inherently difficult to obtain, as sterile males produce no offspring, preventing detection of biased sex-ratios. Indeed, the dual role of *Ovd* in hybrid male sterility and sex-ratio distortion in *D. pseudoobscura* was only detectable because males recover low levels of fertility as they age (*Orr and Irving, 2005*). Although weakly fertile males (producing fewer than five offspring) were removed from the sex-ratio analyses presented here, these males show no evidence for systematically biased sex ratios (*Figure 4—figure supplement 1*).

Our genetic mapping experiments have, however, provided new evidence for the accumulation of cryptic *sex-ratio* drive systems. We mapped a small region of the *D. mauritiana* X that, when introgressed into a naive *D. simulans* genetic background, causes modest segregation distortion resulting in female-biased progeny sex ratios (*Figure 4*). As the *D. mauritiana* X-drive locus does not map to the location of any of the three cryptic drivers known from *D. simulans*, we infer that it may be a new, previously undiscovered drive system in *D. mauritiana*.

Across *D. simulans* and *D. mauritiana*, four cryptic drive systems have been identified so far: two X-drive systems in *D. simulans* (Paris and Durham); one X-drive system in *D. mauritiana* (see above); and one X-drive system found in both species (Winters; see below). We regard this as a minimum for several reasons. First, weak segregation distortion that may be powerful in natural populations can go undetected in laboratory experiments. Second, cryptic drive systems may not be fixed within species, and our genetic mapping experiments have only surveyed genotypes derived from one strain each of *D. mauritiana* and *D. simulans*. Third, no study has yet comprehensively assayed *D. simulans* material introgressed into a *D. mauritiana* genetic background. Finally, some cryptic drive alleles might go to fixation and then simply degenerate because, once fixed (or suppressed), a driver is in a race: either suffer mutational decay or acquire a mutation that confers a new bout of drive. These considerations—and the discovery of multiple alternative cryptic drive systems in closely related species—imply that sex chromosome drive is not infrequent during the history of species divergence (*Jaenike, 2001*).

We have found that the Winters sex-ratio drivers, *MDox* and *Dox*, have migrated between these two species. The two drivers are suppressed by the autosomal suppressor, *Nmy*, which is present in both *D. simulans* and *D. mauritiana* (*Tao et al., 2007a*). The general absence of drive in wild-type genotypes of either species raises one of two possibilities. Either *Nmy* has evolved quickly to

suppress the newly introgressed *MDox* and *Dox* alleles or, alternatively, a suppressing allele of *Nmy* also introgressed between species. We are unable to distinguish these possibilities with the present data, as *Nmy* resides in a chromosomal region dense with complex repetitive sequences that are refractory to genome assembly using short-read data.

The discovery that the *MDox* and *Dox* drivers have moved between species highlights an implicit assumption of the drive theory of the large X-effect—namely, that species evolve in strict allopatry. With gene flow, drive elements (and other selfish genes) have the opportunity to jump species boundaries and undermine divergence in a process analogous to adaptive introgression (*Seehausen et al., 2014*; *Crespi and Nosil, 2013*). The *t*-haplotype has, for instance, introgressed between sub-species of house mouse, *Mus musculus* (*Macaya-Sanz et al., 2011*). Between *D. mauritiana* and *D. simulans*, the $G_{min}$ statistic and the genealogies associated with the *MDox-Dox* introgressed haplotype (*Figure 6*) are agnostic on the direction of introgression. Nonetheless, the finding that a drive element crossed a species boundary has important implications for the drive theory explanation of Haldane's rule and the large X-effect. For *MDox* and *Dox* to introgress between species, three things must be true: (1) neither *MDox* nor *Dox* alleles from the donor species caused male sterility in the recipient species; (2) no X-linked hybrid male sterility factors were so tightly linked to *MDox* and *Dox* as to prevent their eventual escape by recombination into the recipient species genetic background; and (3) any sterility factors located within the introgressed region of the recipient X will have been replaced by foreign alleles. Together, these inferences suggest that a selfish drive system was able to invade a new species by *not* causing male sterility and, for one X-linked region, may have impeded or undone the evolution of hybrid male sterility.

## Materials and methods

**Key resources table**

| Reagent type (species) or resource | Designation | Source or reference | Identifiers | Additional information |
|---|---|---|---|---|
| Genetic reagent (*Drosophila mauritiana*) | mau w[12] | Drosophila species stock center; NCBI SRA | 14021–0241.60; SRX684364; SRX135546 | |
| Genetic reagent (*Drosophila simulans* | sim w[XD1] | this paper | SRR8247551 | obtained from J. Coyne |
| Genetic reagent (*Drosophila mauritiana*) | 2P-1 | this paper | | w[12], P{w[+]=Neneh2}, P{w[+]=4R1} |
| Genetic reagent (*Drosophila mauritiana*) | 2P-3 | this paper | | w[12], P{w[+]=Ophelia1}, P{w[+]=4J1} |
| Genetic reagent (*Drosophila mauritiana*) | 2P-4 | this paper | | w[12], P{w[+]=4J1}, P{w[+]=2A1} |
| Genetic reagent (*Drosophila mauritiana*) | 2P-5a | this paper | | w[12], P{w[+]=2A1}, P{w[+]=ILEA1} |
| Genetic reagent (*Drosophila mauritiana*) | 2P-5b | this paper | | w[12], P{w[+]=2A1}, P{w[+]=2G3} |
| Genetic reagent (*Drosophila mauritiana*) | 2P-6a | this paper | | w[12], P{w[+]=2G3}, P{w[+]=A1} |
| Genetic reagent (*Drosophila mauritiana*) | 2P-6b | this paper | | w[12], P{w[+]=ILEA1}, P{w[+]=A1} |
| Genetic reagent (*Drosophila mauritiana*) | 2P-7 | this paper | | w[12], P{w[+]=A1}, P{w[+]=3L1} |
| Genetic reagent (*Drosophila simulans*) | YFP[175.2] | PMID:28280212 | | pBac{3XP3::EYFP-attP} |
| Genetic reagent (*Drosophila simulans*) | YFP[356.5] | PMID:28280212 | | pBac{3XP3::EYFP-attP} |
| Genetic reagent (*Drosophila simulans*) | YFP[377.31] | PMID:28280212 | | pBac{3XP3::EYFP-attP} |

*Continued on next page*

*Continued*

| Reagent type (species) or resource | Designation | Source or reference | Identifiers | Additional information |
|---|---|---|---|---|
| Genetic reagent (*Drosophila simulans*) | YFP[52.4] | PMID:28280212 | | pBac{3XP3::EYFP-attP} |
| Genetic reagent (*Drosophila simulans*) | YFP[277.1] | PMID:28280212 | | pBac{3XP3::EYFP-attP} |
| Genetic reagent (*Drosophila simulans*) | YFP[926.3] | PMID:28280212 | | pBac{3XP3::EYFP-attP} |
| Genetic reagent (*Drosophila simulans*) | YFP[16.3] | PMID:28280212 | | pBac{3XP3::EYFP-attP} |
| Genetic reagent (*Drosophila simulans*) | YFP[360.1] | PMID:28280212 | | pBac{3XP3::EYFP-attP} |
| Genetic reagent (*Drosophila simulans*) | YFP[433.1] | PMID:28280212 | | pBac{3XP3::EYFP-attP} |
| Genetic reagent (*Drosophila simulans*) | YFP[19.1] | PMID:28280212 | | pBac{3XP3::EYFP-attP} |
| Genetic reagent (*Drosophila simulans*) | YFP[21.4] | PMID:28280212 | | pBac{3XP3::EYFP-attP} |
| Genetic reagent (*Drosophila simulans*) | YFP[458.6] | PMID:28280212 | | pBac{3XP3::EYFP-attP} |
| Sequence-based reagent | Dox_F_1 | this paper | | CGAAATGAGACGCTTCTGTG |
| Sequence-based reagent | Dox_R_1 | this paper | | AACCGATACCG TCGTAGTTGAC |
| Sequence-based reagent | MDox_F_1 | this paper | | CCCATTTTGT CCAAGGTCAC |
| Sequence-based reagent | MDox_R_2 | this paper | | AGTTCCGGTC AAAGTGGTTG |
| Sequence-based reagent | RpS28b_F_1 | this paper | | TGGACAAACC AGTTGTGTGG |
| Sequence-based reagent | RpS28b_R_1 | this paper | | AGGAACTCGA CCTTCACCTG |
| Strain (*Drosophila simulans*) | sim w[501] | PMID:22936249 | 14021–0251.011 | |
| Strain (*Drosophila simulans*) | md06 | NCBI SRA | SRX497551 | |
| Strain (*Drosophila simulans*) | md15 | NCBI SRA | SRX497574 | |
| Strain (*Drosophila simulans*) | md63 | NCBI SRA | SRX497553 | |
| Strain (*Drosophila simulans*) | md73 | NCBI SRA | SRX497563 | |
| Strain (*Drosophila simulans*) | md105 | NCBI SRA | SRX497558 | |
| Strain (*Drosophila simulans*) | md199 | NCBI SRA | SRX497559 | |
| Strain (*Drosophila simulans*) | md221 | NCBI SRA | SRX495510 | |
| Strain (*Drosophila simulans*) | md233 | NCBI SRA | SRX495507 | |
| Strain (*Drosophila simulans*) | md251 | NCBI SRA | SRX497557 | |
| Strain (*Drosophila simulans*) | ns05 | NCBI SRA | SRX497560 | |

*Continued on next page*

*Continued*

| Reagent type (species) or resource | Designation | Source or reference | Identifiers | Additional information |
|---|---|---|---|---|
| Strain (*Drosophila simulans*) | ns33 | NCBI SRA | SRX497575 | |
| Strain (*Drosophila simulans*) | ns39 | NCBI SRA | SRX497562 | |
| Strain (*Drosophila simulans*) | ns40 | NCBI SRA | SRX497556 | |
| Strain (*Drosophila simulans*) | ns50 | NCBI SRA | SRX497571 | |
| Strain (*Drosophila simulans*) | ns67 | NCBI SRA | SRX497565 | |
| Strain (*Drosophila simulans*) | ns78 | NCBI SRA | SRX497573 | |
| Strain (*Drosophila simulans*) | ns79 | NCBI SRA | SRX497576 | |
| Strain (*Drosophila simulans*) | ns113 | NCBI SRA | SRX497572 | |
| Strain (*Drosophila simulans*) | ns137 | NCBI SRA | SRX497561 | |
| Strain (*Drosophila mauritiana*) | r12 | NCBI SRA | SRX135546 | |
| Strain (*Drosophila mauritiana*) | r23 | NCBI SRA | SRX688576 | |
| strain (*Drosophila mauritiana*) | r31 | NCBI SRA | SRX688581 | |
| Strain (*Drosophila mauritiana*) | r32 | NCBI SRA | SRX688583 | |
| Strain (*Drosophila mauritiana*) | r39 | NCBI SRA | SRX688588 | |
| Strain (*Drosophila mauritiana*) | r41 | NCBI SRA | SRX688609 | |
| Strain (*Drosophila mauritiana*) | r44 | NCBI SRA | SRX688610 | |
| Strain (*Drosophila mauritiana*) | r56 | NCBI SRA | SRX688612 | |
| Strain (*Drosophila mauritiana*) | r61 | NCBI SRA | SRX688710 | |
| Strain (*Drosophila mauritiana*) | r8 | NCBI SRA | SRX688712 | |

## *Drosophila* husbandry and genetics

All *Drosophila* crosses and phenotyping were done in parallel in two locations, using standard corn-meal media (Rochester, NY) or minimal cornmeal media (Bloomington, IN) at room temperature (23–25C). We constructed *D. mauritiana* '2P' lines that carry pairs of X-linked *P*-element insertions that contain the mini-*white* transgene ($P[w^+]$) (*True et al., 1996a*) which serve as semi-dominant visible genetic eye-color markers and allow us to distinguish individuals carrying 0, 1 or $2P[w^+]$. These '2P' regions were then introgressed into the *D. simulans* $w^{XD1}$ genetic background through more than 40 generations of repeated backcrossing while following the two $P[w^+]$ insertions (*Figure 1A*). Each 2P introgression line was then bottlenecked through a single female to eliminate segregating variation in the recombination breakpoints flanking the $2P[w^+]$ interval.

We performed meiotic mapping to ascertain the genetic basis of male sterility within each 2P introgression by generating recombinant 1P introgression genotypes (*Figure 1B*). $2P[w^+]$ females were crossed to *D. simulans* strains carrying an X-linked *pBac[eYFP]* transgene (*Stern et al., 2017*) that served as an additional visible marker. Progeny from this cross were scored for recombinant X

chromosomes carrying both *pBac*[*eYFP*] and a single *P*[*w*⁺] (1*P-YFP*). Recombinant 1*P-YFP* chromosomes were generated using *pBac*[*eYFP*] markers both proximal and distal to each 2*P* introgression. Virgin 1*P-YFP* females were individually crossed to *D. simulans w*^XD1 males to initiate 1*P-YFP* strains. Each 1*P-YFP* X chromosome was then assayed for male fertility. At least 10 individual 1*P-YFP* males of each genotype were collected 1–2 days post-eclosion and aged 3–5 days, then placed singly in a vial with three virgin *D. simulans w*^XD1 females. After 7 days, both the male and females were discarded, and all offspring emerging from the vial were counted. Additional 1*P-YFP* males were archived for DNA extraction.

Progeny sex ratios were calculated as the number of female offspring/total number of offspring (% female). Males that sired fewer than five offspring were excluded from sex ratio analyses, as were genotypes with fewer than three males that sired more than four offspring. This resulted in 2538 males and 303 recombinant 1*P*-YFP chromosomes that were used to estimate progeny sex ratios; 210 recombinant 1*P*-YFP genotypes had both progeny sex ratio and sequence data.

## Genotyping recombinant chromosomes by sequencing

We determined the fine-scale genetic architecture of hybrid male sterility within each introgressed region by genotyping recombinant 1*P-YFP* X chromosomes using multiplexed whole-genome sequencing. DNA extraction and library construction followed published methods for high-throughput sequence analysis of a large number of recombinant genotypes (*Andolfatto et al., 2011*; *Peluffo et al., 2015*). Sequence reads were mapped to the reference genome sequence of the *D. mauritiana* stock used for mapping (*mau w*^12) (*Garrigan et al., 2012*), the genome sequence of *sim w*^XD1, and the *D. simulans pBac*[*eYFP*] strains (*Stern et al., 2017*). Ancestry from each parent species was determined by a Hidden Markov Model (HMM) (*Pinero et al., 2017*; *Andolfatto et al., 2011*).

Genotype data and ancestry assignments were inspected for all recombinant 1*P-YFP* introgression genotypes. Genotypes were excluded if there was no segment on the X chromosome identified by the HMM that had either a posterior probability of *D. mauritiana* parentage >0.95 or a posterior probability of *D. simulans* parentage <0.05. Genotypes with segments that had either a posterior probability of *D. mauritiana* parentage >0.95 or a posterior probability of *D. simulans* parentage <0.05 in a region that was not within the parental 2*P* region (i.e. came from a different 2*P* introgression) were inferred to have resulted either from mislabeling or contamination of DNA samples and were excluded from further analyses. 112 genotypes had insufficient sequence data to identify introgressions using the criteria above (or the introgression was too small to be identified). 16 genotypes showed evidence for *D. mauritiana* alleles that did not fall within the parental 2*P* interval. Across the 439 genotypes with sufficiently high-quality sequence data for ancestry assignment, we recovered 64,373 X-linked markers. A subset of 2835 non-redundant markers were retained that delimit the extent of each 1*P-YFP D. mauritiana* segment. No genotype showed evidence for any autosomal *D. mauritiana* alleles (see *Figure 2—figure supplement 1* for exemplars), confirming that our introgression scheme isolated X-linked *D. mauritiana* segments in a pure *D. simulans* autosomal genome.

## Quantitative trait locus analysis

QTL analyses were done in the R/qtl package version 1.36–6. Phenotype means (fertility and progeny sex-ratio) for each introgression genotype and the 2835 non-redundant markers were used as the input data. Mean male fertility was transformed as $\log_{10}$ (N + 1). Because of the large proportion of completely sterile introgression genotypes (*Figure 1—figure supplement 1*), a two-part model (*Broman et al., 2003*) was used to analyze fertility; sex-ratio was analyzed assuming a normal distribution. Significance thresholds were determined using 10,000 permutations of the data.

## Samples and short read alignment

We used genome sequence data from 10 lines of *D. mauritiana*, including nine inbred wild isolates and the genome reference strain, *mau w*^12; 20 lines of *D. simulans*, including 10 inbred wild isolates from Kenya, nine wild isolates from Madagascar, and the reference strain, *sim w*^501; and the reference strain of *D. melanogaster*. The *D. mauritiana* and *D. simulans* sequence data were reported previously (*Garrigan et al., 2012*; *Garrigan et al., 2014*; *Rogers et al., 2014*). SRA accessions for genome sequences are included in the key resources file. The *D. simulans w*^501 and *D. melanogaster*

genome assemblies are available on Flybase (www.flybase.org). We performed short read alignment against the *D. mauritiana* genome assembly (version 2) using the 'aln/sampe' functions of the BWA short read aligner and default settings (*Li and Durbin, 2009*). Reads flanking indels were realigned using the SAMTOOLS software (*Li et al., 2009*). Individual BAM files were merged and sorted with SAMTOOLS.

## Polymorphism and divergence analyses

Both within- and between-population summary statistics were estimated in 10-kb windows using the software package POPBAM (*Garrigan, 2013*). The within population summary statistics include: unbiased nucleotide diversity $\pi$ (*Nei, 1987*); the summary of the folded site frequency spectrum Tajima's D (*Tajima, 1989*); and the unweighted average pairwise value of the $r^2$ measure of linkage disequilibrium, *ZnS*, excluding singletons (*Kelly, 1997*). The between population summary statistics include: two measures of nucleotide divergence between populations, $D_{XY}$, and net divergence, $D_A$ (*Nei, 1987*); the ratio of the minimum between-population nucleotide distance to the average, $G_{min}$ (*Geneva et al., 2015*); and the fixation index, $F_{ST}$ (*Wright, 1951*). From a total of 11,083 scanned 10-kb windows, we only analyzed windows for which at least 50% of aligned sites passed the default quality filters (minimum read coverage 3, minimum rms mapping quality 25, minimum SNP quality 25, minimum map quality 13, minimum base quality 13) in POPBAM, which resulted in a final alignment for 10,443 scanned 10-kb windows. POPBAM output was formatted for use in the R statistical computing environment using the package, POPBAMTools (*Geneva, 2014*). All statistics and data visualization were done in R (*R Development Core Team, 2013*).

## Identification of introgressed regions

We used the $G_{min}$ statistic (*Geneva et al., 2015*) to scan the genome for haplotypes that have recent common ancestry between *D. simulans* and *D. mauritiana*. $G_{min}$ is defined as the ratio of the minimum number of nucleotide differences per aligned site between sequences from different populations to the average number of nucleotide differences per aligned site between populations. The $G_{min}$ statistic was calculated in 10-kb intervals across each major chromosome arm using the same quality filtering criteria used for all other summary statistics. From these values, we estimated the probability of the observed $G_{min}$ under a model of allopatric divergence, conditioned on the divergence time. For each 10-kb genomic interval, the significance of the observed $G_{min}$ value was tested via Monte Carlo coalescent simulation of that 10-kb window with two populations diverging in allopatry with all mutations assumed to be neutral. Simulations were performed using msmove (*Geneva, 2017*), which is based on the coalescent simulation software ms (*Hudson, 2002*), modified to track and report the presence of introgressed genealogies. The arguments of msmove are identical to those of ms and for all simulations we used the following command (msmove 30 10000 t $\theta$ -r $\rho$ 10001 -I 2 10 20 -ej 0.61 1 2). We assumed a population divergence time of $1.21 \times 2_{Nsim}$ generations before the present, in which $N_{sim}$ is the current estimated effective population size of *D. simulans* (*Garrigan et al., 2012*). In the simulations, the observed local value of $D_{XY}$ was used to determine the neutral population mutation rate ($\theta$) for that 10-kb interval. To account for uncertainty in local population recombination rate, for each simulated replicate, a rate was drawn from a normally distributed prior (truncated at zero) with the mean estimated from genetically determined crossover frequencies (*True et al., 1996a*) for that window, and variance equal to the variance of crossover estimates for the entire chromosome arm. The empirical crossover rate estimates were converted from cM to $\rho$ (the population crossover rate, $4N_{sim}c$) by assuming $N_{sim} \approx 10^6$. The effective population sizes of both species were assumed to be equal and constant. For each 10-kb interval, $10^5$ simulated replicates were generated and the probability of the observed $G_{min}$ value was estimated from the simulated cumulative density. To identify putatively introgressed haplotypes, we used a significance threshold of p≤0.001 from the simulations, which yields a proportion of null tests of 0.982 and a false discovery rate of 5%. To infer the full length of any putative introgressions >10-kb, we identified runs of contiguous (or semi-contiguous) 10-kb windows with significant $G_{min}$ values (p≤0.001). We also assessed the distribution of shared derived variants using the four-population test, summarized by Patterson's *D* statistic (*Green et al., 2010*). Variants were generated using POPBAM default parameters and used to calculate Patterson's *D* across chromosome arms using customized perl scripts. For *D* statistic calculations, we assumed the tree structure (((*D. sechellia, D.*

*simulans*), *D. mauritiana*), *D. melanogaster*) for (((P1,P2),P3),O), and used the population frequencies of SNPs to compute probabilistic contributions of individual sites to counts of 'ABBA' and 'BABA' site types (*Green et al., 2010*; *Durand et al., 2011*). Finally, we estimated maximum likelihood phylogenies for each of the putative introgression intervals using RAxML v. 8.1.1 (*Stamatakis, 2014*).

### Genotyping the Winters *sex ratio* genes

We extracted genomic DNA from single male flies using the Qiagen DNeasy Blood and Tissue Kit. The meiotic drive genes of the Winters *sex ratio* system (*Tao et al., 2007a*), *Dox* and *MDox*, were PCR-amplified as previously described (*Kingan et al., 2010*). To assay the presence or absence of the *Dox* and *MDox* gene insertions, the amplicons for the *Dox* and *MDox* regions were digested with the *StyI* and *StuI* restriction enzymes (NEB), respectively. The digests were run on a 1% agarose gel stained with EtBr and the band size was estimated using the GeneRuler 1 kb plus ladder (Thermo Scientific). For both genes, only haplotypes containing the gene insertions have restriction sites as confirmed by samples with known genotypes (*Kingan et al., 2010*).

### Quantitative PCR for *Dox/MDox* expression in fly testes

We assayed expression of the *Dox* and *MDox* genes in testes from *D. simulans* strain MD63 and *D. mauritiana* strain *mau w*[12] using quantitative PCR. Total RNA was extracted from the dissected testes of 5–10 day old flies using the Nucleospin RNA XS kit (Macherey-Nagel, Germany), and cDNA was synthesized with poly dT oligos and random hexamers using Superscript III RT cDNA synthesis kit (Invitrogen, CA). qPCR assays were performed on a BioRad Real-time PCR machine using the cycling conditions: 95° C for 3 mins.; 40 cycles of 95° C for 10 s, 58° C for 30 s, and 72° C for 30 s. The primer sequences used for qPCR are provided in *Supplementary file 3*.

## Acknowledgements

The authors thank Brian Calvi for generously providing laboratory space and Shelby Biel, Ally Shambaugh, and Amanda Meiklejohn for assistance with fertility assays, Cara Brand for calculation of *D. mauritiana* recombination rates, and Peter Andolfatto and Kevin Thornton for sharing *D. simulans* genome sequence data.

## Additional information

### Funding

| Funder | Grant reference number | Author |
| --- | --- | --- |
| National Institute of General Medical Sciences | 1R01OD010548-01A1 | Daniel Garrigan Daven Presgraves |
| University of Nebraska-Lincoln | | Colin Meiklejohn |
| University of Rochester | | Daniel Garrigan Daven Presgraves |
| National Science Foundation | DEB-0839348 | Colin Meiklejohn |
| National Institute of General Medical Sciences | 1R01GM123194-01A1 | Colin Meiklejohn Daven Presgraves |

The funders had no role in study design, data collection and interpretation, or the decision to submit the work for publication.

### Author contributions

Colin D Meiklejohn, Conceptualization, Resources, Formal analysis, Supervision, Funding acquisition, Investigation, Visualization, Methodology, Writing—original draft, Project administration, Writing—review and editing; Emily L Landeen, Conceptualization, Supervision, Investigation, Writing—review and editing; Kathleen E Gordon, Investigation, Writing—review and editing; Thomas Rzatkiewicz, Investigation; Sarah B Kingan, Conceptualization, Software, Formal analysis, Supervision, Investigation, Visualization, Writing—original draft, Writing—review and editing; Anthony J Geneva,

Software, Formal analysis, Methodology, Writing—review and editing; Jeffrey P Vedanayagam, Data curation, Formal analysis, Writing—review and editing; Christina A Muirhead, Formal analysis; Daniel Garrigan, Conceptualization, Resources, Data curation, Software, Formal analysis, Supervision, Funding acquisition, Visualization, Methodology, Writing—original draft, Project administration, Writing—review and editing; David L Stern, Resources, Software, Formal analysis, Investigation, Methodology, Writing—review and editing; Daven C Presgraves, Conceptualization, Resources, Formal analysis, Supervision, Funding acquisition, Investigation, Methodology, Writing—original draft, Project administration, Writing—review and editing

Author ORCIDs

Colin D Meiklejohn https://orcid.org/0000-0003-2708-8316
Daniel Garrigan http://orcid.org/0000-0002-7000-6788
David L Stern http://orcid.org/0000-0002-1847-6483

Decision letter and Author response

Decision letter https://doi.org/10.7554/eLife.35468.048
Author response https://doi.org/10.7554/eLife.35468.049

# Additional files

**Supplementary files**

• Supplementary file 1. $G_{min}$ scan identifies forty-eight interspecific introgressions.
DOI: https://doi.org/10.7554/eLife.35468.029

• Supplementary file 2. Genotype of samples at the *Dox* and *MDox* genes.
DOI: https://doi.org/10.7554/eLife.35468.030

• Supplementary file 3. Primers used in RT-PCR to assay expression of *MDox*, *Dox,* and a control gene (*RpS28b*).
DOI: https://doi.org/10.7554/eLife.35468.031

• Transparent reporting form
DOI: https://doi.org/10.7554/eLife.35468.032

**Data availability**

Sequence data is available via the NCBI Sequence Read Archive (accession number: SRR8247551). Phenotype data have been submitted to Dryad (DOI: https://doi.org/10.5061/dryad.4qn4s47).

The following datasets were generated:

| Author(s) | Year | Dataset title | Dataset URL | Database and Identifier |
|---|---|---|---|---|
| Colin Meiklejohn, Daven Presgraves, David L Stern | 2018 | Sequence data from Gene flow mediates the role of sex chromosome meiotic drive during complex speciation | https://www.ncbi.nlm.nih.gov/sra/?term=SRR8247551 | NCBI Sequence Read Archive, SRR8247551 |
| Meiklejohn CD, Landeen EL, Presgraves DC | 2018 | Gene flow mediates the role of sex chromosome meiotic drive during complex speciation | https://dx.doi.org/10.5061/dryad.4qn4s47 | Dryad Digital Repository, 10.5061/dryad.4qn4s47 |

The following previously published datasets were used:

| Author(s) | Year | Dataset title | Dataset URL | Database and Identifier |
|---|---|---|---|---|
| Garrigan D, Kingan SB, Geneva AJ, Vedanayagam JP, Presgraves DC | 2014 | Drosophila mauritiana genome sequencing | https://www.ncbi.nlm.nih.gov/bioproject/PRJNA158675 | NCBI BioProject, PRJNA158675 |
| Rogers RL, Cridland JM, Shao L, Hu TT, Andolfatto P, Thornton KR | 2015 | Tandem Duplications and the Limits of Natural Selection in Drosophila yakuba and Drosophila simulans | https://www.ncbi.nlm.nih.gov/sra/?term=SRP040290 | NCBI Sequence Read Archive, SRP040290 |

Rogers RL, Cridland JM, Shao L, Hu TT, Andolfatto P, Thornton KR | 2015 | Tandem Duplications and the Limits of Natural Selection in Drosophila yakuba and Drosophila simulans | https://www.ncbi.nlm.nih.gov/sra/?term=SRP029453 | NCBI Sequence Read Archive, SRP029453

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

## Appendix 1

DOI: https://doi.org/10.7554/eLife.35468.033

# Performance of $G_{min}$ - detecting introgression from population genomic data

Using the $G_{min}$ statistic (*Geneva et al., 2015*), we report three findings in the main text regarding historical gene flow between *Drosophila simulans* and *D. mauritiana*:

1. 1.9% of 10-kb windows show evidence for recent introgression between these species;
2. recent introgression is significantly underrepresented on the X chromosome relative to the autosomes; and
3. the lone X-linked region identified as recently introgressed between species contains the previously characterized meiotic drive loci, *Dox* and *MDox*.

In this appendix, we present analyses, simulations, and arguments that support these inferences.

To test $G_{min}$'s power to detect introgression between *D. simulans* and *D. mauritiana*, we used msmove simulations similar to those described in the Methods, assuming a population size $N_e$ of 1,000,000 and 10 generations per year. We simulated divergence followed by gene flow at three times in the past: 400, 4000 and 40,000 years ago. Levels of simulated gene flow were tuned to approximate those observed genome-wide in our data (ms migration probability of 0.008, corresponding to migration occurring in 2–4% of 10-kb windows). Each simulation modeled divergence and gene flow within a 10-kb segment using empirical estimates of population mutation and recombination rate parameters estimated from each 10-kb window in our data. $G_{min}$ was then calculated for each simulated window.

We then used the same procedure described in the Methods to evaluate whether the simulated value of $G_{min}$ for a given window was an outlier by performing 10,000 msmove simulations without gene flow and comparing the $G_{min}$ value from the gene flow simulation to the distribution of $G_{min}$ values from the strictly allopatric simulations. Windows were deemed to be $G_{min}$ outliers if they fell in the lowest 0.001 quantile of the non-gene flow simulated distribution. We repeated these steps 100 times for each 10-kb window at each of the three gene flow time points. The power of $G_{min}$ was determined by measuring the concordance between windows identified as outliers by our procedure and windows that actually contained a simulated gene flow event.

## $G_{min}$ identifies recent introgression

The properties and behavior of the $G_{min}$ statistic have been explored in several previous publications that used coalescent simulations to explore a range of mutation, recombination, and migration parameters. These analyses determined that $G_{min}$ statistical power is robust to variation in recombination and mutation rates (*Geneva et al., 2015*; *Rosenzweig et al., 2016*; *Schrider et al., 2018*). For no set of mutation or recombination parameters considered does $G_{min}$ produce an unacceptably high rate of false positives. $G_{min}$ power is however dependent on the timing of introgression (*Geneva et al., 2015*; *Rosenzweig et al., 2016*; *Schrider et al., 2018*), as shown by simulations that assume levels of gene flow comparable to those observed in our data (*Table 1*). We find that $G_{min}$ detects 86%, 41%, and 2% of simulated gene flow events that occurred 400 years, 4,000 years, and 40,000 years ago, respectively (*Table 1*). While the false positive rate is higher for simulated gene flow that occurred 40,000 years ago, the total number of 10-kb windows with significant $G_{min}$ values is very small; consequently, the total number of false positives is very small as well. We therefore conclude that $G_{min}$ may be unreliable for older introgressions but identifies younger introgressions with high confidence.

**Appendix 1—table 1** $G_{min}$ and power to detect simulated introgression on the X chromosome and autosomes. Numbers in parentheses indicate the standard deviation from 100 replicate simulations

|  | 400 ybp | | 4000 ybp | | 40,000 ybp | |
| --- | --- | --- | --- | --- | --- | --- |
|  | A | X | A | X | A | X |
| Windows with migration (#) | 202.12 (18) | 31.06 (7.2) | 250.65 (16) | 66.48 (10) | 281.15 (20) | 65.22 (8.7) |
| Windows with migration (%) | 2.4% (0.21) | 1.7% (0.4) | 3% (0.19) | 3.7% (0.56) | 3.4% (0.24) | 3.6% (0.48) |
| Significant $G_{min}$ windows (#) | 179.08 (17) | 28.5 (6.4) | 111.44 (9.8) | 27.02 (4.7) | 15.87 (4.4) | 2.4 (1.3) |
| Significant $G_{min}$ windows (%) | 2.2% (0.2) | 1.6% (0.35) | 1.3% (0.12) | 1.5% (0.26) | 0.19% (0.052) | 0.13% (0.07) |
| True positive rate | 96% (1.5) | 95% (3.9) | 94% (2.4) | 93% (5.1) | 45% (11) | 30% (34) |
| False postive rate | 3.7% (1.5) | 4.8% (3.9) | 5.7% (2.4) | 6.8% (5.1) | 55% (11) | 70% (34) |
| Migration Events Detected | 85% (3.2) | 88% (7.5) | 42% (2.9) | 38% (5.1) | 2.6% (1) | 1.2% (1.4) |

DOI: https://doi.org/10.7554/eLife.35468.034

## Comparing introgression on the X *versus* autosomes

X-linked loci have (generally) smaller effective population sizes than autosomal loci and hence lower levels of nucleotide polymorphism. We tested if the systematically lower observed polymorphism on the X can explain its lower levels of introgression detected by $G_{min}$ using coalescent simulations, resampling, and additional statistical analyses of our empirical results.

First, we note that $G_{min}$ statistical significance for each 10-kb window was determined by Monte Carlo simulation of neutral genealogies derived from two populations diverging in allopatry (no gene flow). As these simulations used estimates of $\theta$ and $\rho$ drawn from each 10-kb window, they necessarily incorporate systematic differences between the X and autosomes in these parameters when generating *P*-values. Second, the simulations presented in *Table 1* show no significant difference in power (true positive rate or proportion of migration events detected) between X-linked and autosomal windows. Third, using a resampling approach, we generated 10,000 'X-matched autosome' datasets, each drawn randomly from autosomal 10-kb windows, that closely matched the distribution of polymorphism among true X-linked windows and tallied the number of significant $G_{min}$ windows (*Figure 1*). For 10,000 'X-matched autosome' datasets matching X-linked polymorphism in *D. simulans*, no dataset had as few or fewer significant $G_{min}$ windows than the actual X-linked data (p<0.0001); for 10,000 'X-matched autosome' datasets matching polymorphism in *D. mauritiana*, only one dataset had as few significant $G_{min}$ windows as the actual X-linked data (p=0.0001). These findings suggest that the observed paucity of introgressions on the X chromosome cannot be explained simply by its lower levels of polymorphism.

Our data do reveal negative correlations between the $G_{min}$ *P*-value and polymorphism within *D. simulans* (Spearman's $\rho = -0.22$, p<0.0001) and within *D. mauritiana* ($\rho = -0.38$, p<0.0001). Importantly, this correlation is driven by variation in *P*-values among the large majority of non-significant windows (see *Figure 5—figure supplement 2*). However, significant $G_{min}$ windows on average do have different levels of polymorphism than non-significant ones. In *D. simulans*, significant $G_{min}$ windows have less polymorphism than non-significant windows, whereas, in *D. mauritiana*, significant $G_{min}$ windows have *more* polymorphism than non-significant windows (*Figure 5—figure supplement 2*; Wilcoxon test p<0.0001 for both species). The observation that significant $G_{min}$ windows have elevated polymorphism in *D. mauritiana* may reflect the direction of gene flow: the presence of

foreign alleles will tend to elevate diversity in the receiving (*D. mauritiana*) population but not the donor population (*D. simulans*).

## Patterson's *D* statistic may not be appropriate for X-autosome comparisons

In the main text, we report that Patterson's *D* for the genome (*D* = 0.0812, combining all chromosomes), is smaller than that for the the X chromosome (*D* = 0.1054, excluding the 130-kb *Dox* region). For all autosomes combined, *D* = 0.077, yielding a X/A ratio of *D* = 1.361. Superficially, these values could imply the possibility of *more* introgression on the X than the autosomes and would therefore seem to contradict the $G_{min}$ results which suggest the opposite. We suggest however that Patterson's *D* statistic may be inappropriate for simple X *versus* autosome comparisons and that discrepancies between Patterson's *D* and scans for introgression are not unique to our study.

In the case of constant population size, the expected value of *D* is inversely related to $N_e$ (*Green et al., 2010*; *Durand et al., 2011*). As a result, under most circumstances, a larger value of *D* is expected for the X chromosome even if all else— including the degree of introgression— is constant. To illustrate the point, we calculated expected *D* using standard assumptions for our *Drosophila* species and obtain values similar to those estimated from the data. The simplifying assumptions for all calculations are:

- constant $N_e$ = 1,000,000 for all species
- 10 generations per year
- a three-species polytomy of *D. mauritiana*, *D. simulans*, *D. sechellia*
- a speciation time = 250,000 years (2,500,000 gens) in the past
- a single pulse of gene flow occurring 50,000 years (500,000 gens) in the past
- introgression probability, *f* = 0.05

Using these assumptions and Equation 5 from (*Durand et al., 2011*) yields *E*[*D*]=0.072. Taking this value as a plausible autosomal expectation for *D*, we considered three different $N_e$ values for the X, while holding all other parameters constant. *Table 2* provides expectations for *D* on the X chromosome and X/A ratios of *D*.

**Appendix 1—table 2.** X chromosome, and X/A ratio, for expectation of Patterson's *D*.

| X/A ratio of $N_e$ | Rationale | *E*[*D*] | X/A ratio of *D* |
|---|---|---|---|
| 0.75 | 1:1 sex ratio, random mating, *etc.* | 0.094 | 1.309 |
| 0.656 | Observed X/A nucleotide diversity in *D. mauritiana* | 0.106 | 1.479 |
| 0.778 | Observed X/A nucleotide diversity in *D. simulans* | 0.091 | 1.265 |

DOI: https://doi.org/10.7554/eLife.35468.035

For all three cases, the X/A ratio of *E*[*D*] is greater than one and comparable to the ratio estimated from our data (X/A ratio = 1.361) despite no difference in assumptions about the amount or timing of gene flow between the X and autosomes.

Notably, discrepancies between Patterson's *D* and focused introgression scans are not unique to our study. Green et al. (*Green et al., 2010*) developed the *D* statistic and estimated *D* between Neanderthal and non-African humans for all 23 chromosomes (their Supplementary Table S47). Between Asians and Neanderthals, *D* is 2.3-fold higher for the X chromosome than the mean of the 22 autosomes (Table S47), which would seem to imply a greater rate of introgression on the X. Later work by the same group (*Sankararaman et al., 2014*; *Sankararaman et al., 2016*) scanned genomes for introgression using relative sequence distances and haplotype length as criteria and found a significant dearth of introgression on the X relative to the autosomes (X/A introgression ~20%). Thus, paralleling our results, Patterson's *D* between Asians and Neanderthals implies excess gene flow on the X, whereas the genomic scan implies the opposite. As expectations for Patterson's *D* on the

X *versus* the autosomes are confounded by effective population size, it seems imprudent to draw strong conclusions about relative gene flow from the *D* statistic alone.

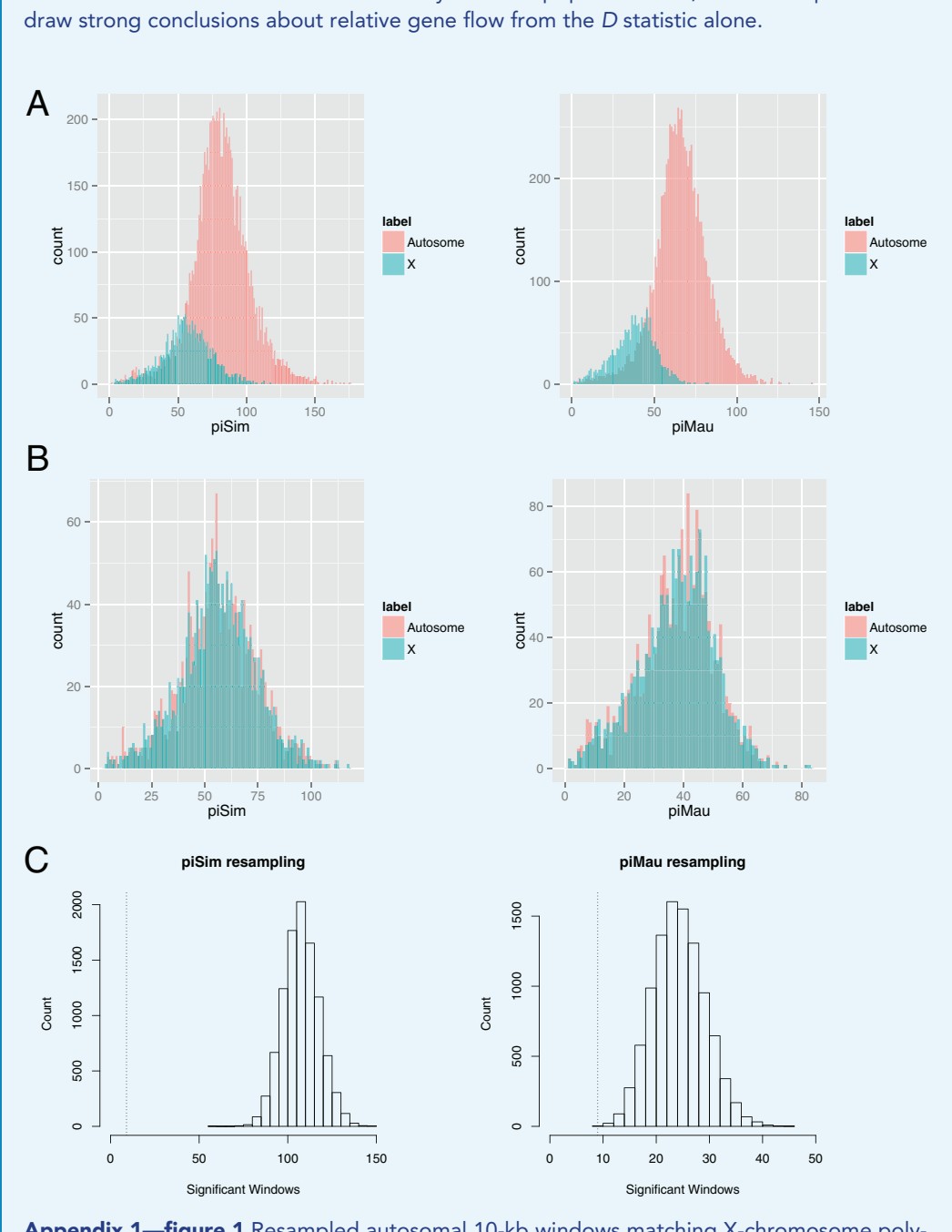

**Appendix 1—figure 1** Resampled autosomal 10-kb windows matching X-chromosome polymorphism. (**A**) Distributions of polymorphism within 10-kb windows for the X chromosome and autosomes in *D. simulans* and *D. mauritiana*. (**B**) Exemplar resampled autosomal data sets matching X-chromosome polymorphism for *D. simulans* and *D. mauritiana*. (**C**) Distribution of the number of resampled windows with significant $G_{min}$ values across 10,000 replicate resampled data sets. Vertical dotted lines indicate the observed number of significant X-linked windows in each species.

DOI: https://doi.org/10.7554/eLife.35468.036

