## [Decision Letter]

Thank you for submitting your article "Gene flow mediates the role of sex chromosome meiotic drive during complex speciation" for consideration by *eLife*. Your article has been reviewed by three peer reviewers, and the evaluation has been overseen by a Reviewing Editor and Diethard Tautz as the Senior Editor. The reviewers have opted to remain anonymous.

The reviewers have discussed the reviews with one another and the Reviewing Editor has drafted this decision to help you prepare a revised submission. As you will see from the comments below, all three reviewers appreciated the impressive amount of work and thought the findings were interesting and novel. However, they had some concerns, notably that (1) *G*_min_ is a relatively untested method and its biases could potentially confound interpretation of the results. In that regard, they suggested either evaluating its performance by simulation or using it alongside standard methods such as the D statistic and checking that the conclusions remain unchanged. (2) The reviewers pointed out that both the use of *G*_min_ and the mapping analysis is insufficiently detailed (or not described at all); please make sure to add this information in your revision. Other suggestions are included below, in individual comments.

Reviewer #1:

Meiklejohn et al., use a clever design to map to moderate resolution several male-sterile hybrid incompatibilities between *D. mauritiana* and *D. simulans*. They also find several large regions that are tolerant to introgression on the X, including a known drive element that has undergone introgression between species in natural populations. The experimental approach is novel and the findings interesting, but I have some concerns outlined below:

1a) Analyses of gene flow. The observed pattern of introgression around a drive locus is very interesting, but I had several concerns about the analysis and would like to see a more comprehensive analysis of the gene flow signal reported. First, I was unsure whether *G*_min_ is the most sensitive and relevant statistic for this test, in part because it seems that it will only detect introgressing segments that are not fixed. If true, this seems like it could contribute to the frequency difference in introgressed segments observed on the X and autosomes. Second, it would be helpful to get a sense of the expected performance of *G*_min_ under a number of conditions such a recombination and mutation rate variation, particularly given the observation that regions identified have systematically smaller numerators and denominators. I was also unsure how the setup of the simulation procedure could influence the results.

1b) Winters sex ratio drive system results. I was wondering if the authors could expand on these observations and analyses. First, I was wondering if the moderate *G*_min_ reduction here suggests that this introgression might be old (?) but the fact that it is identified by *G*_min_ and the authors' other analyses suggests that it is not fixed. Or is a ~2-fold reduction in *G*_min_ expected for very recent introgression? It would also be informative in interpreting the results if the authors added permutations on what the probability of overlap of a 130 kb introgressed region with a known drive system on the X is by chance; I expect it would be low. Furthermore, I was confused about the argument that these regions had experienced selective sweeps in both species. I believe for *simulans* this is in reference to previous work, but evidence for *mauritiana* appears to be in reference to lower Dxy within *mauritiana* in this region. I would recommend a more formal analysis of this or tempering the language somewhat.

2) Possible limitations of the mapping design and analysis. If I am understanding the setup correctly, one possible limitation is that there may be many incompatible segments within the introgressed segments, but only a subset of recombination events that combine the white eye and fluorescent markers can be detected. Thus, the authors rely on the QTL mapping to try to localize these incompatible regions, but there may be several or many within each region. To me this is suggested by the reported relationship between introgressed region length and fertility, as well as qualitatively by the high background in the mapping results. In order to tease this apart, the authors could add mixture proportion of the X as a covariate in their mapping analyses to understand how much individual regions explain sterility/fertility above the length of the introgressed segment. I think either conclusion is quite interesting. I do think dealing with the variation in mixture proportion between lines is necessary because it appears quite substantial from Figure 2 and thus could be viewed as substantial population structure in the mapping population.

3) Lack of important detail in analyses. The manuscript lacks important detail necessary to understand and interpret the results. I noticed this particularly in the description of mapping of fertility/sterility and in the Monte Carlo simulations. For mapping, it appears that rQTL was used but I did not find details of this in the Methods (approach, phenotype distribution assumed, covariates, permutation approach, etc.). For the Monte Carlo simulations, which are key to interpreting the *G*_min_ results, I did not see details on how the simulations were done. i.e. what program was used or was an in-house program used, how large were the segments simulated, how was the appropriateness of null demographic models evaluated, among many other details.

It would be very interesting to add some simulations of expected dynamics of drive loci in this hybridization scenario or make more explicit reference to the findings of previous modeling if this is outside of the scope.

Reviewer #2:

This manuscript by Meiklejohn and colleagues provides a comprehensive body of work involving genetic crosses and genomic analyses to address fundamental questions regarding the role of meiotic drive in speciation. In particular, this study brings together introgression analysis of the X chromosome and population genomic analyses between *Drosophila simulans* and *D. mauritiana*. This is a staggering amount of work, the methods and analyses appear sound, the conclusions are both robust and novel, and the manuscript is presented in a clear and concise manner.

The authors first use a 2P-based introgression system to move tiled sections of the X chromosome from *D. mauritiana* into *D. simulans* through >40 backcross generations, which allows them to coarsely map hybrid male sterility loci. They then use a 1P-YFP based method to perform high resolution mapping of these loci. Among the fertile introgressions generated during this study, they uncovered a cryptic drive system in *D. mauritiana*. This drive system is novel and does not correspond to any of the three known systems in *D. simulans*. Among the sterile introgressions, Meiklejohn and colleagues detect four small regions that cause hybrid male sterility. None of these sterile introgressions appear to show signs of meiotic drive. Together, the authors conclude that there is ample evidence for cryptic drive between species, but none of the sterile introgressions show any association with drive elements.

1) Here, it is probably worth pointing out that if a history of drive led to the evolution of any of the hybrid male sterile loci, this would be undetectable. Sterile introgression males, by definition, produce few or no kids and would not allow robust detection of biased sex ratios. Such males may, however, sometimes prove to be very weakly fertile. Progeny data from such very weakly fertile males (< five kids), however, are discarded in the current analysis. Along these lines, it may be worth separately describing the progeny sex ratios in very weakly fertile males generated in otherwise sterile introgressions.

The population genomic analyses uncover 47 genomic regions of introgression between these species located on autosomes, and only one on the X chromosome. This X-linked region corresponds to a very small region that includes genes involved in a known *D. simulans* drive system (*Dox/ MDox*). This region of low divergence is associated with a known drive system but not with any of the hybrid male sterile genes. This suggests that gene flow across the two species, perhaps mediated by the selfish spread of this driver, prevented the evolution of hybrid incompatibility genes in this region of the X-chromosome. These results indicate that selfish drive systems may not only promote the evolution of hybrid sterility between species as has been shown in other studies but may also act against the evolution of reproductive isolation in certain cases.

2) Here, it is worth noting that the X chromosomes of most strains in *D. simulans* or *D. mauritiana* do not distort within species. If the Winters system has indeed invaded across species, then this may suggest that its suppressors of drive may also have accompanied the driving locus. In this context, it may be worth noting whether the region on 3R that corresponds to the known suppressor *Nmy* also show signs of introgression.

Reviewer #3:

This manuscript has three large components. First Meiklejohn et al., map the genetic basis of hybrid sterility in a recently diverged species pair of *Drosophila*. They use high resolution mapping to identify the genetic basis of hybrid sterility in *simulans/mauritian*a hybrid males (carrying the *mauritiana* X chromosome). Sometimes the term high resolution mapping gets overhyped; not in this case, these authors really mean it. They used multiple mapping methods to validate their results (Figure 1, Figure 2 and Figure 3). The find six (!) separable elements involved in sterility. The second portion of the manuscript uses the experimental introgressions (one of the approaches the authors used to map the genetic basis of hybrid sterility) and find that some introgressions lead to cryptic X-chromosome drive in *D. mauritiana*. These results are important because there are very few known drive systems for which the molecular basis is known. These are important experiments and they try to elucidate the reasons of why X-chromosomes are commonly involved in isolation. Moreover, we know little about how meiotic drive systems affect genome divergence (but see below). These two components of the manuscript are nothing short of stunning and frankly it's some of the best evolutionary genetics I have read in the last 12 months. I have no major comments for this section.

The third component of the manuscript is a population genomics approach to identify regions from the X-chromosome that have crossed species boundaries. The authors use the *G*_min_ metric to detect introgressions in the X-chromosome. This metric is slightly problematic though. *G*_min_ was originally proposed by Geneva et al., (2015). After studying the paper, I found it has no real estimates of sensitivity of the method; the closest it gets is a comparison between *G*_min_ and Fst. The latter is not a proper metric to detect gene flow and its shortcomings have been discussed at length (e.g., Noor and Bennett, 2010; Guerrero and Hahn, 2017 among many others). A related issue is that the method implemented in POPBAM uses windows to assess the existence of gene flow. This is a limited approach to infer the presence of small introgressions: if the X-linked alleles are strongly selected against, one would expect smaller haplotypes which means windows in the X-chromosome have less power than windows in the autosomes to detect introgression. The authors also use this window-based approach to calculate the size of the haplotype around *Dox* and infer that its size is 130kb (the upper panel of Figure 6 is not very informative). Then the authors infer selective sweeps in this region. This 'haplotype' shows reduced polymorphism and lower interspecific divergence than the rest of the genome. This result is interesting, but I am left to wonder how does *G*_min_ perform in instances of low polymorphism (i.e., what is the power of the metric). Schrider et al., (bioRxiv, Figure 1) provides an estimate of the performance of *G*_min_ (but compared to their own method, so not very useful here) and conclude that *G*_min_ can lead to false positives.

Overall, the second half of the manuscript is weaker than the first, but I think that if the authors can demonstrate that *G*_min_ performs well to detect interspecific gene flow with high sensitivity and specificity, I would be convinced of the results. I would suggest two possible solutions here. The authors could (1) simulate genomes with different levels of introgression and determine whether the metric performs well with the level of divergence observed between *simulans* and *mauritiana* (Geneva et al., 2015 has some forward simulations that really do not address this issue) or (2) the authors use an additional method to validate their results with POPBAM.

Two additional notes on this topic. First, I am a little surprised there is no mention of the possibility of incomplete lineage sorting (ILS) and its possible involvement with meiotic drive. For example, if the haplotype around *Dox* and *MDox* is truly 130kb, ILS is a truly unlikely explanation. If the signal is caused by smaller segments of shared ancestry that get collated into a large window, then it is less likely. (This comment is related to my concern about the use of *G*_min_ to detect the size of an introgression.)

An additional suggestion for the authors. There is serious amount of important work in this manuscript. I had to read the piece over half a dozen times and the connection between sterility, drive, and gene exchange never crystallized. A couple of statements summarizing the results and stating the connection between sections would solve this issue and make this piece much more enjoyable.

[Editors' note: further revisions were requested prior to acceptance, as described below.]

Thank you for resubmitting your work entitled "Gene flow mediates the role of sex chromosome meiotic drive during complex speciation" for further consideration at *eLife*. Your revised article has been favorably evaluated by Diethard Tautz (Senior Editor), a Reviewing Editor, and two of the original three reviewers.

Everyone appreciated your revisions, but the reviewers had a couple of additional suggestions that we would like you to address before publication, most of which involve slight revisions to the text or clarifications to be added. We leave it to your discretion if you want to take up suggestion #1 of reviewer 1, or further discuss the D statistic results and the concordance with other lines of evidence in the text.

Reviewer #1:

The authors made a number of the suggested changes from the first round of review and I believe that the paper is much improved. I had a couple of additional questions/concerns regarding the added analyses and details included in this version of the manuscript.

1) I appreciate the authors adding D-statistic analyses in addition to *G*_min_ analyses. One question arising from these analyses was that the estimate of Patterson's D for the X chromosome was not lower than the rest of the genome, and this seems to conflict with other results. This made me concerned that the observations about fewer *G*_min_ identified windows on the X were due to power differences or some other issue and not due to lower introgression. Since D is not a direct of measure of mixture proportion, something like an F4-ratio could be used and I would be reassured if this resulted in lower estimates of introgression on the X. It would also be a helpful reality check if it could be shown that mixture proportions are estimated to be lower in regions with mapped hybrid sterility loci from this study (and possibly higher in regions estimated to be permeable to introgression based on mapping).

2) Data quality questions. I was concerned about what seemed like adhoc data quality evaluation in the fifth paragraph of the Materials and methods section. It seems like visual inspection could be replaced with a formal threshold (e.g. n kb with high posterior probability *mauritiana* ancestry, or a particular coverage threshold). How was evidence of contamination evaluated and how many lines did this impact? It would also be helpful to know many genotypes were excluded due to these criteria.

I also became concerned about reference bias issues and signals of introgression in reading the added details of the POPBAM analyses. The minimum coverage threshold of 3 reads seemed much too low to me based on experience with this kind of data. Low coverage can exacerbate issues with reference bias and given that mapping was to the *mauritiana* reference rather than an outgroup, could potential impact inferences of gene flow. I do not have an intuition about how this could impact *G*_min_ analyses but have found it to have an impact on D-statistic type analyses.

Comment on response to reviewers: In response to my previous comments about *G*_min_ the authors note "There is little evidence for recent parallel, hard selective sweeps from population genomic data for these species to date[…]" I am not sure that the population genetic observations relating to non-admixed models the authors detail here are directly relevant, as the dynamics after admixture can be quite different particularly with selection (both negative and positive). As before, I would prefer analyses of local ancestry that were sensitive to fixed regions as I think it would give more insight into the history of admixture, but do not think this impacts the main results of the paper which remain exciting.

Reviewer #3:

The points related to the genetic mapping have been addressed.

I still have a quibble regarding the detection of introgression. The authors use *G*_min_ to detect introgressions and admit the metric is best suited to detect introgression in instances of recent (I'd argue very recent) introgression. In this new version they add calculations of the D-statistic on genomic windows to obtain an independent confirmation of their results.

I have reservations about a few statements in the manuscript though.

Since *G*_min_ is dependent on dmin, its power will depend on the amount of polymorphism on a window. Since the magnitude of variation is different between autosomes and X-chromosomes (subsection “Population genomics of speciation history”); I am not sure *G*_min_ is a good metric to compare the magnitude of introgression between X chromosomes and autosomes (as stated in the sixth paragraph).

The newly added analysis, D calculated on 10kb genomic-windows suffers from similar issues as *G*_min_. Simon et al., (2014) describes the statistical properties of the metric in detail.

I lean to think that differences in π or dmin can fully explain the autosome/X ratio in number of introgressed windows of 47:1 but I think is worth including the caveat.

I am convinced the DOX alleles have crossed species boundaries, but I still think the language of the manuscript needs a little bit of clean up.

---

## [Author Response]

Reviewer #1:Meiklejohn et al., use a clever design to map to moderate resolution several male-sterile hybrid incompatibilities between D. mauritiana and D. simulans. They also find several large regions that are tolerant to introgression on the X, including a known drive element that has undergone introgression between species in natural populations. The experimental approach is novel and the findings interesting, but I have some concerns outlined below:1a) Analyses of gene flow. The observed pattern of introgression around a drive locus is very interesting, but I had several concerns about the analysis and would like to see a more comprehensive analysis of the gene flow signal reported. First, I was unsure whether G_min_ is the most sensitive and relevant statistic for this test, in part because it seems that it will only detect introgressing segments that are not fixed.

To clarify, *G*_min_ can detect genomic segments of historical introgression in which the introgressed haplotype is (1) segregating in both populations, or (2) fixed in one population and segregating in the other. For example, our ~130-kb X-linked introgression is fixed in *D. mauritiana* but segregating in *D. simulans*.

The only introgressions that cannot be detected are those fixed in both species. We have empirical evidence that such reciprocal monophyly (at least for 10-kb windows) is rare: under reciprocal monophyly, expected *G*_min_→1; but there are no windows where *G*_min_ =1, and 95% of *G*_min_ values are <0.85 (see Figure 5). Furthermore, we argue below that the expected number of introgressions fixed in both species by drift should be negligible and those fixed in both species by positive selection should be detectable (due to information at flanking sites; see below).

If true, this seems like it could contribute to the frequency difference in introgressed segments observed on the X and autosomes. Second, it would be helpful to get a sense of the expected performance of G_min_ under a number of conditions such a recombination and mutation rate variation, particularly given the observation that regions identified have systematically smaller numerators and denominators. I was also unsure how the setup of the simulation procedure could influence the results.

The publication describing *G*_min_ included analyses of the statistic's performance over a range of both recombination and mutation rates (Geneva et al.,2015). These analyses showed that *G*_min_ is largely insensitive to variation in these parameters.

We suggest that our inability to detect introgressed segments that are fixed in both species is unlikely to account for the observed deficit of X-linked introgressions. We consider the fixation of introgressions under (1) the standard neutral model (SNM) and (2) a standard selective sweep (SSM) model.

Under SNM assumptions, shorter transit times for neutral X-linked alleles destined to fixation are expected based on the smaller effective population size of the X chromosome. As we are interested in haplotypes that become fixed in *both* populations, the quantity of interest is the expected time to reciprocal monophyly (i.e., coalescence within each of the two populations). For an autosomal locus, Hudson and Coyne, 2002 showed that the probability of reciprocal monophyly reaches 0.95 only after 8.7*N*_e_ generations— this is >7-fold older than the inferred split time for our species. (Note that the Hudson-Coyne calculations assume a random partitioning of an ancestral population of size *N* into two descendant populations both also of size *N*; this history is different than the introgression of a haplotype from one population into another via gene flow. Nevertheless, the theory clearly predicts longer conditional times to fixation than expected for a single population, i.e., >4*N* generations.) For neutral introgressions, then, genealogies that are reciprocally monophyletic constitute a negligible fraction of sites on the X chromosome and the autosomes.

Under a model of parallel selective sweeps, reciprocal monophyly can in principle occur much faster than the neutral case. However, we believe the parallel sweeps model will have limited impact in reducing the efficacy of *G*_min_, for three reasons:

1) Parallel, hard selective sweeps are required to fix the introgressed haplotype in both species. There is little evidence for *recent* parallel, hard selective sweeps from population genomic data for these species to date.

2) The complete parallel sweeps will only produce reciprocal monophyly for the sequences corresponding to the overlap between the two fixed haplotypes. Most sweeps result in relatively short tracts that are identical-by-descent (IBD) within species, and the length of the overlap of the two IBD tracts will be smaller still.

3) Even under conditions that produce reciprocal monophyly, we expect to be able to detect introgression using *G*_min_*at flanking sites*. The reason is that flanking sites will enter the receiving population with the introgression; hitchhike to non-trivial population frequencies; but then get recombined away from the beneficial introgressed mutation(s). (Note that this is the same process that causes elevated LD and excess rare variants at flanking sites in the wake a classic hard sweep.) While the beneficial mutation goes to fixation, flanking sites that “escape” the sweep will segregate as polymorphic introgressed haplotypes and will therefore be detectable by *G*_min_.

Finally, we note that an earlier study that identified candidate introgressed regions using only a single sequence from *D. simulans* and *D. mauritiana* also found a deficit of X-linked introgressions (Garrigan et al., 2012).

1b) Winters sex ratio drive system results. I was wondering if the authors could expand on these observations and analyses. First, I was wondering if the moderate G_min_ reduction here suggests that this introgression might be old (?) but the fact that it is identified by G_min_ and the authors' other analyses suggests that it is not fixed. Or is a ~2-fold reduction in G_min_ expected for very recent introgression?

We predict no simple relationship between the age of an introgressed segment and quantitative reduction in *G*_min_, as this statistic is determined by both the similarity of between-species haplotypes and by the frequency of the introgressed haplotype in both populations through the average *D*_XY_. Two considerations suggest that the introgression at *Dox-MDox* occurred recently: first, at the *Dox* locus we find no fixed differences within the introgressed haplotype between *D. simulans* and *D. mauritiana*. Second, *G*_min_ has limited power to detect older introgressions (see more details on this below).

It would also be informative in interpreting the results if the authors added permutations on what the probability of overlap of a 130 kb introgressed region with a known drive system on the X is by chance; I expect it would be low.

We have approached this question in two ways. First, at a cutoff of p < 0.001 (FDR = 0.05) we find that nine of 1,842 10-kb windows on the X chromosome have a significant *G*_min_ value. In 10,000 random permutations of the data, no permutation resulted in a significant *G*_min_ value for both 10-kb windows containing *Dox* and *MDox,* suggesting the probability of observing this result by chance is less than 0.0001. Second, these nine significant 10kb windows are all located within a single 130-kb region on the X chromosome; hence our inference that they together represent a single introgression event. *MDox* and *Dox* are physically separated by five 10-kb windows,so that there are seven 130-kb segments that include both loci. As there are 1,830 10-kb windows on the X chromosome, the probability that a randomly placed 130-kb window includes both *MDox* and *Dox* is 7/1830 = 0.004. Both of these approaches suggest that the probability of overlap of a 130-kb introgressed region with a known drive system on the X by chance is indeed low. We have included this second estimate in the revised manuscript (Discussion section).

There is a second known X-linked drive system in *D. simulans,* known as the Paris system. This system comprises two loci separated by ~160 kb that are both required for segregation distortion and thus could not be included within a 130-kb introgressed segment.

Furthermore, I was confused about the argument that these regions had experienced selective sweeps in both species. I believe for simulans this is in reference to previous work, but evidence for mauritiana appears to be in reference to lower Dxy within mauritiana in this region. I would recommend a more formal analysis of this or tempering the language somewhat.

Two previously published studies have presented formal analyses indicating a very strong, recent, hard selective sweep at the *Dox/MDox* locus in *D. mauritiana* (Nolte et al., 2013; Garrigan et al., 2014), and a third study documented evidence of recent positive selection at *Dox* and *MDox* in *D. simulans* (Kingan, Garrigan and Hartl, 2010). Garrigan et al., 2014 used the same genome sequences for *D. mauritiana* analyzed here, while Kingan et al., 2010 detected positive selection in a set of *D. simulans* isolates that are distinct from the sequences analyzed in this manuscript. We have updated the text of the manuscript to more clearly highlight this past work (Results section).

2) Possible limitations of the mapping design and analysis. If I am understanding the setup correctly, one possible limitation is that there may be many incompatible segments within the introgressed segments, but only a subset of recombination events that combine the white eye and fluorescent markers can be detected.

The mapping approach we used is able to detect all recombination events between the two *P[w*^+^] markers. For each 2*P* interval, we used *eYFP* markers located outside the two *P[w*^+^] markers, and we identified recombination events via the loss of one *P[w*^+^] marker (through changes in eye color from red or dark orange to light orange) and the gain of fluorescence. The utility of the *eYFP* marker was to trap specific recombination breakpoints and allow repeated measurement of fertility among males that all carry the same recombinant segment.

However, our approach *is* limited in its ability to identify the effects of hybrid sterility loci located in the middle of each 2*P* segment (perhaps this was the reviewer's point). As each recombinant genotype carries one *P[w*^+^] marker, sterility factors closely linked to the *P[w*^+^] may mask the effects of additional sterility factors within the 2*P* segment further from the *P[w*^+^]. This is most clearly seen for the interval 2*P*-6, where we detect complete sterility associated with both the left and right *P[w*^+^] markers, and thus cannot determine whether additional sterility factors reside in the middle of the interval. This limitation is discussed in the manuscript (Materials and methods section).

Thus, the authors rely on the QTL mapping to try to localize these incompatible regions, but there may be several or many within each region. To me this is suggested by the reported relationship between introgressed region length and fertility, as well as qualitatively by the high background in the mapping results. In order to tease this apart, the authors could add mixture proportion of the X as a covariate in their mapping analyses to understand how much individual regions explain sterility/fertility above the length of the introgressed segment. I think either conclusion is quite interesting. I do think dealing with the variation in mixture proportion between lines is necessary because it appears quite substantial from Figure 2 and thus could be viewed as substantial population structure in the mapping population.

It is true that the length of *D. mauritiana* segments varies extensively between recombinant introgression genotypes, although it is not clear to us how this could be considered equivalent to be population structure. We have re-examined the relationship between introgression length and fertility and conclude that there is less evidence for weak sterilizing factors distributed across the X chromosome than we first suspected. The previously reported negative correlation between introgression length and fertility (Pearson's *r* = -0.22, p<0.0001; Spearman's *r* = -0.31, p<0.0001) is almost entirely attributable to long, completely sterile introgressions. When we include only the 264 genotypes with a mean fertility >0, the correlation between length and fertility is greatly reduced (*r* = -0.13, *P* = 0.04; *r* = -0.14, *P* = 0.02). When we include only the 210 genotypes with sufficient fertility to be included in our sex-ratio analyses, this correlation disappears entirely (*r* = -0.05, *P* = 0.49; *r* = -0.03, *P* = 0.75). We therefore conclude that the effect of introgression length is largely the result of long introgressions being more likely to carry large-effect *D. mauritiana* alleles that significantly reduce male fertility.

This conclusion is supported by the analyses including introgression length suggested by the reviewer. For some of the QTL models we have implemented, it is not possible to use introgression length as a covariate in the R/qtl package. However, for the models where it is possible, we have added analyses that include introgression length as a covariate (see Figure 3—figure supplement 2). The results are largely consistent with analyses that do not include this covariate; we conclude that identification of major sterility regions via QTL analysis is not seriously confounded by the quantitative effects of introgression length on male fertility. The revised Results section now refer only to partial correlation analyses between introgression length, fertility, and progeny sex-ratio.

3) Lack of important detail in analyses. The manuscript lacks important detail necessary to understand and interpret the results. I noticed this particularly in the description of mapping of fertility/sterility and in the Monte Carlo simulations. For mapping, it appears that rQTL was used but I did not find details of this in the Methods (approach, phenotype distribution assumed, covariates, permutation approach, etc). For the Monte Carlo simulations, which are key to interpreting the G_min_ results, I did not see details on how the simulations were done. i.e. what program was used or was an in-house program used, how large were the segments simulated, how was the appropriateness of null demographic models evaluated, among many other details.

We have updated the Materials and methods section to include these details that were previously omitted (QTL analysis and the Monte Carlo simulations).

It would be very interesting to add some simulations of expected dynamics of drive loci in this hybridization scenario or make more explicit reference to the findings of previous modeling if this is outside of the scope.

We agree that this is an interesting question, but it is beyond the scope of our (already lengthy) manuscript. Unfortunately, while theory exists for adaptive introgression in the context of linked genetic incompatibilities (Uecker et al., 2015), we’re unaware of comparable treatments of selfish introgression of drive loci in the context of linked genetic incompatibilities.

Reviewer #2:This manuscript by Meiklejohn and colleagues provides a comprehensive body of work involving genetic crosses and genomic analyses to address fundamental questions regarding the role of meiotic drive in speciation. In particular, this study brings together introgression analysis of the X chromosome and population genomic analyses between Drosophila simulans and D. mauritiana. This is a staggering amount of work, the methods and analyses appear sound, the conclusions are both robust and novel, and the manuscript is presented in a clear and concise manner.The authors first use a 2P-based introgression system to move tiled sections of the X chromosome from D. mauritiana into D. simulans through >40 backcross generations, which allows them to coarsely map hybrid male sterility loci. They then use a 1P-YFP based method to perform high resolution mapping of these loci. Among the fertile introgressions generated during this study, they uncovered a cryptic drive system in D. mauritiana. This drive system is novel and does not correspond to any of the three known systems in D. simulans. Among the sterile introgressions, Meiklejohn and colleagues detect four small regions that cause hybrid male sterility. None of these sterile introgressions appear to show signs of meiotic drive. Together, the authors conclude that there is ample evidence for cryptic drive between species, but none of the sterile introgressions show any association with drive elements.1) Here, it is probably worth pointing out that if a history of drive led to the evolution of any of the hybrid male sterile loci, this would be undetectable. Sterile introgression males, by definition, produce few or no kids and would not allow robust detection of biased sex ratios. Such males may, however, sometimes prove to be very weakly fertile. Progeny data from such very weakly fertile males (< five kids), however, are discarded in the current analysis. Along these lines, it may be worth separately describing the progeny sex ratios in very weakly fertile males generated in otherwise sterile introgressions.

This important point has now been highlighted in the Discussion section. The progeny sex ratios from all males, including sub-fertile males are now presented in Figure 4—figure supplement 1 and Figure 4—figure supplement 2. In short, we detect no evidence for systematically female-biased progeny sex-ratios among sub-fertile males

The population genomic analyses uncover 47 genomic regions of introgression between these species located on autosomes, and only one on the X chromosome. This X-linked region corresponds to a very small region that includes genes involved in a known D. simulans drive system (Dox/ MDox). This region of low divergence is associated with a known drive system but not with any of the hybrid male sterile genes. This suggests that gene flow across the two species, perhaps mediated by the selfish spread of this driver, prevented the evolution of hybrid incompatibility genes in this region of the X-chromosome. These results indicate that selfish drive systems may not only promote the evolution of hybrid sterility between species as has been shown in other studies but may also act against the evolution of reproductive isolation in certain cases.2) Here, it is worth noting that the X chromosomes of most strains in D. simulans or D. mauritiana do not distort within species. If the Winters system has indeed invaded across species, then this may suggest that its suppressors of drive may also have accompanied the driving locus. In this context, it may be worth noting whether the region on 3R that corresponds to the known suppressor Nmy also show signs of introgression.

Unfortunately, the region on 3R containing *Nmy* is surrounded by complex repeat sequences that preclude its assembly using short-read sequence data. As a consequence, the *Nmy* locus is missing from our assembly and alignments, and we cannot at the moment address whether *Nmy* has co-introgressed with *MDox/Dox*. This is however an interesting point we intend to return to in future work.

Reviewer #3:This manuscript has three large components. First Meiklejohn et al. map the genetic basis of hybrid sterility in a recently diverged species pair of Drosophila. They use high resolution mapping to identify the genetic basis of hybrid sterility in simulans/mauritiana hybrid males (carrying the mauritiana X chromosome). Sometimes the term high resolution mapping gets overhyped; not in this case, these authors really mean it. They used multiple mapping methods to validate their results (Figure 1, Figure 2 and Figure 3). The find six (!) separable elements involved in sterility. The second portion of the manuscript uses the experimental introgressions (one of the approaches the authors used to map the genetic basis of hybrid sterility) and find that some introgressions lead to cryptic X-chromosome drive in D. mauritiana. These results are important because there are very few known drive systems for which the molecular basis is known. These are important experiments and they try to elucidate the reasons of why X-chromosomes are commonly involved in isolation. Moreover, we know little about how meiotic drive systems affect genome divergence (but see below). These two components of the manuscript are nothing short of stunning and frankly it's some of the best evolutionary genetics I have read in the last 12 months. I have no major comments for this section.

We appreciate these generous comments from the reviewer.

The third component of the manuscript is a population genomics approach to identify regions from the X-chromosome that have crossed species boundaries. The authors use the G_min_ metric to detect introgressions in the X-chromosome. This metric is slightly problematic though. G_min_ was originally proposed by Geneva et al., (2015). After studying the paper, I found it has no real estimates of sensitivity of the method; the closest it gets is a comparison between G_min_ and Fst. The latter is not a proper metric to detect gene flow and its shortcomings have been discussed at length (e.g., Noor and Bennett 2010; Guerrero and Hahn, 2017 among many others). A related issue is that the method implemented in POPBAM uses windows to assess the existence of gene flow. This is a limited approach to infer the presence of small introgressions: if the X-linked alleles are strongly selected against, one would expect smaller haplotypes which means windows in the X-chromosome have less power than windows in the autosomes to detect introgression. The authors also use this window-based approach to calculate the size of the haplotype around Dox and infer that its size is 130kb (the upper panel of Figure 6 is not very informative). Then the authors infer selective sweeps in this region. This 'haplotype' shows reduced polymorphism and lower interspecific divergence than the rest of the genome. This result is interesting, but I am left to wonder how does G_min_ perform in instances of low polymorphism (i.e., what is the power of the metric). Schrider et al., (bioRxiv, Figure 1) provides an estimate of the performance of G_min_ (but compared to their own method, so not very useful here) and conclude that G_min_ can lead to false positives.

We agree that a plausible cause of the dearth of X-linked introgressions is that X-linked foreign alleles are more strongly selected against, leading to smaller introgressions on the X that are more difficult to detect (see Materials and methods section). However, we consider this to be an interesting biological phenomenon, rather than a limitation of the *G*_min_ statistic per se.

Simulations presented both in Geneva et al., 2015 (https://doi.org/10.1371/journal.pone.0118621) and Schrider et al., 2018 (https://doi.org/10.1371/journal.pgen.1007341) indicate that *G*_min_ has the greatest power to detect recent introgressions. Our interpretation of Figure 1 in Schrider et al., 2018 is not that *G*_min_ has a high rate of false positives, but rather that it has a high rate of false negatives for all but the most recent of introgressions. We therefore conclude that our analysis has likely missed older introgressions, but that we can be reasonably confident about the introgressions it did identify.

The relationship between polymorphism and the probability of identifying introgression by *G*_min_ is complex. However, two considerations suggest that the reduced polymorphism at the *MDox/Dox* locus should not lead to a spurious inference of introgression. First, simulations presented in Geneva et al., 2015 indicate that *G*_min_'s false negative rate (proportion of truly introgressed segments missed by *G*_min_; 1 – sensitivity) increases with decreasing polymorphism, while the false positive rate (proportion of segments identified by *G*_min_ that did not truly introgress; 1 – specificity) is insensitive to levels of polymorphism. This suggests that we should expect a greater rate of false negatives in regions with low polymorphism, and that the identification of introgressed *MDox/Dox* alleles is conservative.

Second, there is a significant negative correlation between the *G*_min_*P-*value and polymorphism within both *D. simulans* (Spearman's *r*=-0.22, p<0.0001) and *D. mauritiana (r*=-0.38, p<0.0001), indicating that windows with higher polymorphism tend to have lower *G*_min_ values, although this correlation is largely driven by the large majority of non-significant windows (Figure 5—figure supplement 2). However, 10-kb windows with significant *G*_min_ values have lower levels of polymorphism in *D. simulans* than non-significant windows, while significant windows have *higher* levels of polymorphism in *D. mauritiana* than non-significant windows (Figure 5—figure supplement 2). One interpretation of this pattern is that windows identified as significant by *G*_min_ have levels of polymorphism similar to that in the other species (*D. simulans* harbors more polymorphism than *D. mauritiana*), which we think is consistent with these windows carrying lineages derived from the other species. Altogether, these considerations suggest that our inference of introgression at *MDox/Dox* should be robust to the low levels of polymorphism at this locus.

Overall, the second half of the manuscript is weaker than the first, but I think that if the authors can demonstrate that G_min_ performs well to detect interspecific gene flow with high sensitivity and specificity, I would be convinced of the results. I would suggest two possible solutions here. The authors could (1) simulate genomes with different levels of introgression and determine whether the metric performs well with the level of divergence observed between simulans and mauritiana (Geneva et al., 2015 has some forward simulations that really do not address this issue) or (2) the authors use an additional method to validate their results with POPBAM.

As mentioned above, the properties and behavior of *G*_min_ have been explored in at least three publications, Geneva *et al.* 2015, Schrider et al., 2018, and Rosenzweig et al., 2016. Geneva et al., 2015 studied its behavior, sensitivity, and specificity using coalescent simulations that explore a range of migration, mutation and recombination parameters. Rosenzweig et al., 2016 similarly investigated the power of *G*_min_ for varying simulated levels of migration and migration time. Schrider et al., 2018 compared the performance *G*_min_ with their method that integrates multiple sequence features and summary statistics, including *G*_min_. As mentioned above, Schrider et al., 2018 showed that their method has increased power (fewer false negatives) relative to *G*_min_, but not that *G*_min_ suffers from a high rate of false positives. Finally, we controlled the genome-wide false discovery rate to 5%, corresponding to *P*-values <0.001. Thus, we feel that our implementation of *G*_min_ to detect regions of introgression has been adequately documented to justify these analyses.

Nonetheless, to show that our inference that the *MDox/Dox* region introgressed between species is robust to the method used to identify introgression, we have implemented the four-population (ABBA-BABA) test, summarized by Patterson's *D* statistic. Within the 130-kb *MDox/Dox* region we find that a large excess of derived sites is shared between *D. simulans* and *D. mauritiana* to the exclusion of *D. sechellia* relative to both the rest of the X chromosome and to the autosomes. We conclude that this approach also supports the recent movement of alleles at this locus between *D. simulans* and *D. mauritiana*. These results are presented in the revised manuscript (Discussion section).

Two additional notes on this topic. First, I am a little surprised there is no mention of the possibility of incomplete lineage sorting (ILS) and its possible involvement with meiotic drive. For example, if the haplotype around Dox and MDox is truly 130kb, ILS is a truly unlikely explanation. If the signal is caused by smaller segments of shared ancestry that get collated into a large window, then it is less likely. (This comment is related to my concern about the use of G_min_ to detect the size of an introgression.)

We never seriously entertained the possibility that ILS could explain the signal of introgression at *MDox/Dox*, for multiple reasons including those outlined by the reviewer. There are many ancestral polymorphisms segregating in these two species (see Table 4), consistent with incomplete lineage sorting (ILS). But these ancestral polymorphisms segregate as SNPs interspersed with new, derived SNPs that have accumulated since the species diverged. They do not persist as long, intact ancestral haplotypes. Therefore, ILS does not produce significantly reduced *G*_min_ estimates for 10-kb windows. The inferred introgressed region comprises 9 nearly contiguous 10-kb windows with *G*_min_*P*-values ranging from <0.0001 to a maximum of 0.0007 (Supplementary file 1). These are windows that show significantly low sequence distances between subsets of haplotypes in the two species that cannot be accommodated by an isolation model (and hence ILS). Put another way, the region has a dearth of derived, species specific SNPs in these haplotypes. Finally, we note that a history of introgression is supported by additional four-population test (Patterson’s *D*; see above).

An additional suggestion for the authors. There is serious amount of important work in this manuscript. I had to read the piece over half a dozen times and the connection between sterility, drive, and gene exchange never crystallized. A couple of statements summarizing the results and stating the connection between sections would solve this issue and make this piece much more enjoyable.

We appreciate this suggestion, and we have tried to highlight these connections in the revised manuscript (Abstract and Results section).

[Editors' note: further revisions were requested prior to acceptance, as described below.]

Reviewer #1:Thank you for resubmitting your work entitled "Gene flow mediates the role of sex chromosome meiotic drive during complex speciation" for further consideration at eLife. Your revised article has been favorably evaluated by Diethard Tautz (Senior Editor), a Reviewing Editor, and two of the original three reviewers.Everyone appreciated your revisions, but the reviewers had a couple of additional suggestions that we would like you to address before publication, most of which involve slight revisions to the text or clarifications to be added. We leave it to your discretion if you want to take up suggestion #1 of reviewer 1, or further discuss the D statistic results and the concordance with other lines of evidence in the text.

We appreciate the reviewers' time and care in considering our revisions. In the interests of brevity and readability, we have added an appendix to our manuscript that describes additional simulations and calculations that support our inferences of introgression gleaned from *G*_min_ and the *D* statistic. Below we address the reviewer comments, making reference to the Appendix where appropriate.

The authors made a number of the suggested changes from the first round of review and I believe that the paper is much improved. I had a couple of additional questions/concerns regarding the added analyses and details included in this version of the manuscript.1) I appreciate the authors adding D-statistic analyses in addition to G_min_ analyses. One question arising from these analyses was that the estimate of Patterson's D for the X chromosome was not lower than the rest of the genome, and this seems to conflict with other results. This made me concerned that the observations about fewer G_min_ identified windows on the X were due to power differences or some other issue and not due to lower introgression. Since D is not a direct of measure of mixture proportion, something like an F4-ratio could be used and I would be reassured if this resulted in lower estimates of introgression on the X. It would also be a helpful reality check if it could be shown that mixture proportions are estimated to be lower in regions with mapped hybrid sterility loci from this study (and possibly higher in regions estimated to be permeable to introgression based on mapping).

The value of Patterson's *D* statistic is influenced by effective population size, and thus, even in the absence of chromosomal differences in introgression, the X chromosome is expected to show greater values of *D* than the autosomes. We demonstrate this property in the Appendix and refer to previously published studies using human and Neanderthal genome sequences that also show a deficit of X-linked introgression by distance methods (analogous to *G*_min_) but higher values of *D* on the X chromosome than the autosomes.

2) Data quality questions. I was concerned about what seemed like adhoc data quality evaluation in the fifth paragraph of the Materials and methods section. It seems like visual inspection could be replaced with a formal threshold (e.g. n kb with high posterior probability mauritiana ancestry, or a particular coverage threshold). How was evidence of contamination evaluated and how many lines did this impact? It would also be helpful to know many genotypes were excluded due to these criteria.

The perception of an *ad hoc* approach was due to an overly simplified description of the actual data filtering process. We have revised the manuscript to include more specific details regarding this data filtering step. The revised Materials and methods section now reads:

"Genotype data and ancestry assignments were inspected for all recombinant 1*P-YFP* introgression genotypes. Genotypes were excluded if there was no segment on the X chromosome identified by the HMM that had either a posterior probability of *D. mauritiana* parentage > 0.95 or a posterior probability of *D. simulans* parentage < 0.05. Genotypes with segments that had either a posterior probability of *D. mauritiana* parentage >0.95 or a posterior probability of *D. simulan*s parentage < 0.05 in a region that was not within the parental 2*P* region (i.e. came from a different 2*P* introgression) were inferred to have resulted either from mislabeling or contamination of DNA samples and were excluded from further analyses. 112 genotypes had insufficient sequence data to identify introgressions using the criteria above (or the introgression was too small to be identified). 16 genotypes showed evidence for *D. mauritiana* alleles that did not fall within the parental 2*P* interval."

We have included data files and figures that visualize ancestry assignments for all genotypes in our data submission to Dryad, including those that were excluded from further analyses.

I also became concerned about reference bias issues and signals of introgression in reading the added details of the POPBAM analyses. The minimum coverage threshold of 3 reads seemed much too low to me based on experience with this kind of data. Low coverage can exacerbate issues with reference bias and given that mapping was to the mauritiana reference rather than an outgroup, could potential impact inferences of gene flow. I do not have an intuition about how this could impact G_min_ analyses but have found it to have an impact on D-statistic type analyses.

We do not expect that this is a problem for our analyses. Reference bias should cause divergent *D. simulans* sequences that fail to map to the *D. mauritiana* genome assembly to be excluded from our analyses; missing these divergent alleles will decrease average *D_XY_* and thus increase *G*_min_ and reduce our power. Thus, our choice of parameters should be conservative with respect to identifying introgressions.

Comment on response to reviewers: In response to my previous comments about G_min_ the authors note "There is little evidence for recent parallel, hard selective sweeps from population genomic data for these species to date[…]" I am not sure that the population genetic observations relating to non-admixed models the authors detail here are directly relevant, as the dynamics after admixture can be quite different particularly with selection (both negative and positive). As before, I would prefer analyses of local ancestry that were sensitive to fixed regions as I think it would give more insight into the history of admixture, but do not think this impacts the main results of the paper which remain exciting.

We feel that these issues, while interesting, are beyond the scope of the present manuscript.

Reviewer #3:The points related to the genetic mapping have been addressed.I still have a quibble regarding the detection of introgression. The authors use G_min_ to detect introgressions and admit the metric is best suited to detect introgression in instances of recent (I'd argue very recent) introgression. In this new version they add calculations of the D-statistic on genomic windows to obtain an independent confirmation of their results.I have reservations about a few statements in the manuscript though.Since G_min_ is dependent on dmin, its power will depend on the amount of polymorphism on a window. Since the magnitude of variation is different between autosomes and X-chromosomes (subsection “Population genomics of speciation history”); I am not sure G_min_ is a good metric to compare the magnitude of introgression between X chromosomes and autosomes (as stated in the sixth paragraph).The newly added analysis, D calculated on 10kb genomic-windows suffers from similar issues as G_min_. Simon et al., (2014) describes the statistical properties of the metric in detail.I lean to think that differences in π or dmin can fully explain the autosome/X ratio in number of introgressed windows of 47:1 but I think is worth including the caveat.I am convinced the DOX alleles have crossed species boundaries, but I still think the language of the manuscript needs a little bit of clean up.

In the Appendix, we present calculations and simulations that indicate that the power of *G*_min_ is not significantly different for the X chromosome and the autosomes, and that the lower polymorphism of the X chromosome is unlikely to explain the observed deficit of X-linked introgression. We concur that, unlike *G*_min_, the *D* statistic is inappropriate for comparisons between the X and autosomes, and we outline this reasoning in the Appendix. Nonetheless, we feel the *D* statistic has value for demonstrating a history of past introgression between these two species, and for confirming that the *Dox/MDox* region has a genealogical history that is distinct from the rest of the X chromosome.